

# Technical note: New approach for the determination of $N_2$ fixation rates by coupling a membrane equilibrator to a mass spectrometer on voluntary observing ships

Sören Iwe[1], Oliver Schmale[1], Bernd Schneider[1]

[1]Department of Marine Chemistry, Leibniz Institute for Baltic Sea Research Warnemünde (IOW), Rostock, 18119, Germany

*Correspondence to*: Sören Iwe (soeren.iwe@io-warnemuende.de)

**Abstract.** Nitrogen fixation by cyanobacteria plays an important role in the eutrophication of the Baltic Sea, since it promotes biomass production in the absence of dissolved inorganic nitrogen (DIN). Its contribution to the N budget is of the same order of magnitude as the combined sum of riverine and airborne DIN input, varying between 310 kt-N/yr and 792 kt-

N/yr. The vast range is due to interannual variability, significant uncertainties in the various techniques used to determine $N_2$ fixation and in extrapolating local studies to entire basins. To overcome some of the limitations we introduce a new approach using a Gas Equilibrium – Membrane-Inlet Mass Spectrometer (GE-MIMS). A membrane contactor (Liquicel) is utilized to establish gas phase equilibrium for atmospheric gases dissolved in seawater. The mole fractions for $N_2$, Ar and $O_2$ in the gas phase are determined continuously by mass spectrometry and yield the concentration of these gases by multiplication with

the total pressure and the respective solubility constants. The results from laboratory tests show that the accuracy (deviation from expected values): $N_2$: 0.20 %, Ar: 0.70 %, $O_2$: 0.20 % and the precision (2 times the absolute standard deviation) $N_2$: 0.05 %, Ar: 0.14 %, $O_2$: 0.11 % is sufficient enough to detect and quantify nitrogen fixation. The *e*-folding equilibration time is 4.8 min for $N_2$, 3.0 min for Ar and 3.2 min for $O_2$. The GE-MIMS is designed for deployment on a voluntary observing ship (VOS), enabling repeated transects along the same route and providing high temporal and spatial resolution data.

Therefore, the method is suitable for offering large-scale records of the surface water $N_2$ depletion and of Ar to account for the air-sea gas exchange. Additional $O_2$ measurements will be utilized to estimate the net community production (NCP) triggered by $N_2$ fixation.

## 1 Introduction

Measuring the levels of dissolved gases such as $CO_2$, $O_2$ and $N_2$ in seawater is an established approach to investigate the

biogeochemical processes associated with the production or decomposition of organic matter (OM). It is particularly well suited for monitoring coastal waters and semi-enclosed seas such as the Baltic Sea, where excessive nutrient inputs from river water and atmospheric deposition often lead to increased OM production (eutrophication). In the Baltic Sea, nitrogen fixation by cyanobacteria utilizing molecular nitrogen dissolved in surface water in the absence of dissolved inorganic nitrogen (DIN) can further amplify OM production.



These biogeochemical processes are linked to changes in the concentrations of dissolved atmospheric gases, and specifically
      the consumption and production of $CO_2$ and $O_2$, respectively, during net community production (NCP) and $N_2$ depletion
      through nitrogen fixation. An increase in NCP can in turn lead to $O_2$ depletion and the formation of $H_2S$ in deeper water
      layers, due to the microbial oxidation of OM. Denitrification at the sedimentary or pelagic oxic/anoxic interface can promote
      $N_2$ production. The efficiencies of the latter processes in the Baltic Sea are favored by the years of stagnation in the deeper

water layers of its central basins (Schneider et al., 2017).

      The concentrations of gases dissolved in seawater is often determined based on the analysis of air at equilibrium with the
      seawater, a state reached using various equilibrators and with frequently direct contact between the gas and water phases.
      Measurements of discrete samples can be made using the conventional headspace method whereas bubble- or shower-type
      equilibrators or membrane equilibrators are used for continuous measurements (e.g. Gülzow et al., 2011; Jahangir et al.,

2012; Tortell et al., 2015; Schmale et al., 2019). For example, bubble-type equilibrators in combination with infrared
      spectroscopy have been used for continuous large scale surface-water $p CO_2$ (partial pressure of $CO_2$) records (Schneider et
      al., 2009; Schneider et al., 2014). The $p CO2$ data were obtained using a fully automated measurement system deployed on a
      voluntary observation ship (VOS) traveling 4–5 times per week over a distance of > 1000 km across the entire central Baltic
      Sea. Changes in total $CO_2$ ($C_T$) were determined from the $pCO_2$ measurements. This method takes into account $CO_2$ gas

exchange with the atmosphere and the formation of dissolved organic carbon to calculate seasonal $C_T$ depletion, in turn
      facilitating estimates of the net production of particulate organic matter (POM) and thus NCP. The presence of NCP during
      mid-summer, when no DIN is available, implies the occurrence of $N_2$ fixation, which can be quantified on the basis of the
      mean C/N ratio of POM in mid-summer in the central Baltic Sea.

      However, as this estimate of $N_2$ fixation is indirect, with many associated uncertainties, Schmale et al. (2019) developed an

alternative approach in which a spray-type equilibrator is coupled with a mass spectrometer to obtain direct and continuous
      measurements of the $N_2$ concentration. Application of this method during a research cruise in mid-summer in the Baltic Sea
      revealed a distinct $N_2$ depletion in the surface water, attributed to $N_2$ fixation since it coincided with a clear draw-down of $C_T$
      in the absence of DIN. These findings demonstrated that mass spectrometry is sufficiently sensitive to detect the surface-
      water $N_2$ depletion caused by $N_2$ fixation. However, since the measurements were performed at different times in different

regions, it was not possible to derive $N_2$ fixation rates. Also, established methods for quantifying $N_2$ fixation are based on
      excess phosphorus consumption ($PO_4$ approach, Rahm et al., 2000), the total nitrogen increase in surface water (TN
      approach, Larsson et al., 2001; Eggert et al., 2015), or $^{15}N_2$ incubation ($^{15}N$ approach, Montoya et al. 1996), all of which have
      their limitations. The most significant shortcoming common to all three is related to their reliance on the analysis of discrete
      samples. Due to the patchiness of cyanobacterial blooms, measurements based on discrete samples can introduce significant

uncertainties when extrapolating or interpolating the data in space and time. The wide range of $N_2$ fixation estimates (310 -
      792 kt-N/yr, Wasmund et al., 2005; Rolff et al., 2007) for the Baltic Proper reflects interannual variability but also these
      serious methodological uncertainties.



A mass spectrometry technique for the analysis of gases dissolved in seawater was introduced in the early 1960s by Hoch and Kok (1963). Membrane inlet mass spectrometry (MIMS) uses a membrane to separate a continuous flow of water from the gas side (headspace), with the direct connection of this setup to a mass spectrometer (MS). Due to the continuous pumping of the gas into the MS, low pressure in the headspace leads to the flow of the dissolved gases across the membrane into the headspace. As a result, a steady state is established in which the concentrations of the individual gases in the headspace depend on the concentrations of the dissolved gases and the MS pumping rate. A requirement of this technique is the calibration of the system with seawater containing defined concentrations of the gases of interest. Due to the short response time MIMS facilitates the detection of fast changes in the dissolved gas concentrations, by reducing the ratio between the volume of the headspace and the gas flow into the MS. MIMS has been employed in the determination of NCP on the basis of $O_2$ measurements and the $O_2/Ar$ saturation ratio to describe the physically shaped $O_2$ background concentration (Kaiser et al., 2005; Nemcek et al., 2008; Tortell et al., 2015).

The present study introduces a modification of MIMS, gas equilibrium-membrane-inlet mass spectrometry (GE-MIMS), with the most significant difference being the establishment of a gas-phase equilibrium (Cassar et al., 2009; Mächler et al., 2012; Manning et al., 2016). The latter is maintained by the removal of only minor amounts of gas from the headspace, without creating a negative pressure. After calibration of the MS with standard gases, the equilibrium partial pressures, linked to the concentrations of the dissolved gases by their solubility constants, can be determined. Compared to bubble-/shower-type equilibrators, membrane equilibrators do not require ventilation and are therefore closed systems. This ensures that the partial pressures in the gas phase are truly at equilibrium with the dissolved gases rather than merely at steady state (Schneider et al., 2007).

Our newly developed GE-MIMS system is designed to track eutrophication in the Baltic Sea, with a particular focus on the surface concentration of $N_2$ ($N_2$ fixation). The results will be supported by concurrent measurements of the $O_2$ concentration, which provides complementary information about NCP, while determinations of the Ar concentration facilitate a reconstruction of abiotic conditions and allow for the determination of the gas exchange of N2 and O2.

The specific aims of this study were the following:

(1) determination of the completeness of the equilibration process and of the equilibration (response) time using commercially available membrane contactors;

(2) determination of the precision/accuracy of the gas analysis by mass spectrometry and estimates of the limits of detection for biogenic changes in $O_2$ and $N_2$ concentrations;

(3) the prospective deployment of the GE-MIMS system on a VOS to better capture the importance of the nitrogen fixation for the Baltic Sea nitrogen budget;



## 2 Measuring device

### 2.1 The membrane equilibrator

The fundamental principle of GE-MIMS is an equilibration of the partial pressures on the water side and gas side
(comparable to a headspace) which are separated by a gas-permeable membrane.

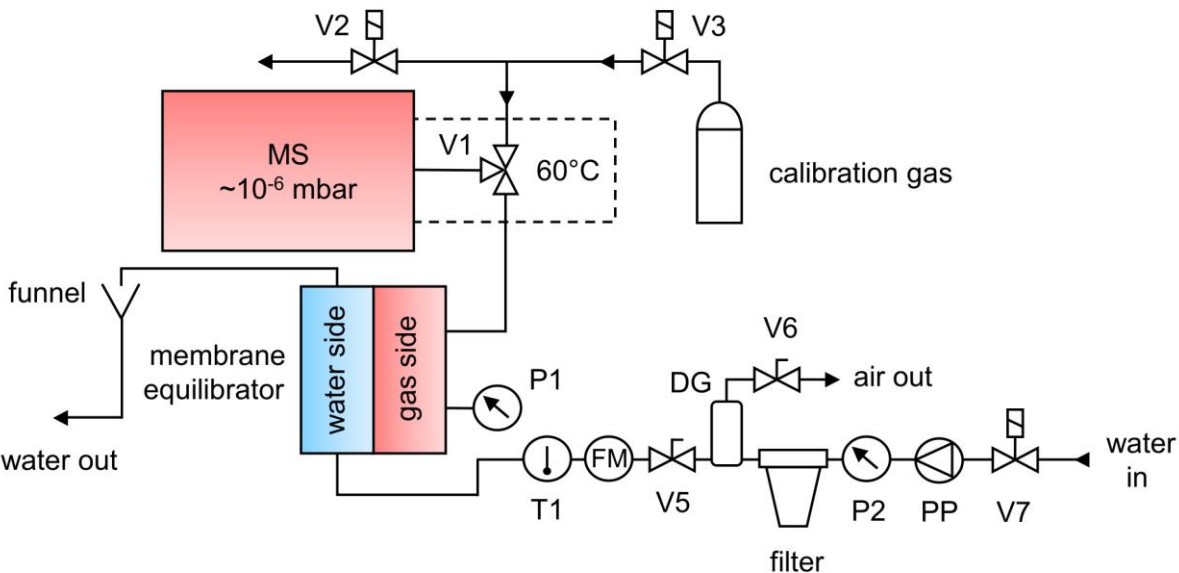

**Figure 1: Schematic diagram of the gas equilibrium – membrane-inlet mass spectrometry (GE-MIMS) system.**

To ensure that the measurement system is appropriate for use at sea, it should first be tested in a laboratory setup designed as
follows: A continuous flow of water is generated with the aid of a peristaltic pump. Seawater samples should be filtered prior
to their exposure to the membrane to prevent its clogging with particles. This can be accomplished by installing a 10" 
polypropylene filter cartridge with a 5-µm pore size. A pressure gauge (P2) installed ahead of the filter indicates when the 
filter must be changed. Air trapped in the filter cartridge can be removed with the help of two valves (V5 and V6) and a

degassing cylinder directly behind it. If laboratory experiments are conducted using distilled or tap water, filtration and the
associated air removal are unnecessary.

The filter cartridge used in this study was the Liqui-Cel mini-module cartridge ($1.8 \times 8.75$, model G541), containing porous
hollow fibers made of gas-permeable hydrophobic polypropylene. The fibers are compactly arranged in a polycarbonate
housing and provide a total membrane surface area of 0.9 m². The water volume (70 mL) of the cartridge has almost no

effect on the equilibration time, but the latter significantly decreases with higher water flow rates, typically 1–3 L/min



(see Sect. 3.2). Due to the relatively large gas room (140 mL) and the high permeability of the membrane, the continuous flow of gas into the MS (6 µL/min) caused only a negligible disturbance of the equilibration process (see Appendix B).

A membrane cartridge (PermSelect 1m$^2$) with dense silicone (PDMS) hollow fibers was tested for its utility in avoiding clogging of the hollow fiber pores by particles, but it was not used due to excessive water vapor condensation in the

headspace, which posed a risk to the MS.

Accurate water temperature measurements in the GE-MIMS system are provided by a temperature probe (T1, PT100, precision: 0.01°C) located at the inlet of the membrane. Total gas tension is recorded by connecting a pressure sensor (P1) (SEN 3276, Kobold, precision: 2 mbar) to the gas side of the membrane cartridge. A capillary coupling the gas side with the MS is connected via V1.

An important aspect of the water flow system is to ensure that the outlet is not positioned below the membrane outlet. This is to prevent a suction effect due to gravity leading to a negative pressure, which will ultimately affect the equilibration within the membrane. In the GE-MIMS system, this is accomplished by the use of an outlet with an adjustable height and connected to a funnel.

## 2.2 Mass spectrometry for $N_2$, Ar and $O_2$

A commercially available quadrupole mass spectrometer (QMS, GAM2000, InProcess Instruments) was used to analyze the gas composition on the gas side of the membrane equilibrator. A high vacuum ($10^{-6}$ mbar) is generated within the MS through the combined use of a membrane pump and a turbomolecular pump. The sample gas is introduced to the ion source through a deactivated fused silica capillary (length: 3 m, diameter: 50 µm) connected to a 2-position valve (V1). To enhance MS signal stability, these parts are housed within a heated enclosure maintained at a constant temperature of 60°C.

Using V1, either the gas side of the equilibrator or the calibration line is connected to the MS by a capillary. It is crucial that the two capillaries have identical properties (1.5 m, Ø 50 µm) in order to maintain a constant internal pressure within the MS. The capillaries' dimensions limit the gas flow rate to approximately 6 µL/min according to the modified Hagen-Poiseuille equation (Cassar et al., 2009). This rate is consistent with a measured transfer time from capillary inlet to detection of approximately 80 s.

During calibration, valve V3 connects the MS with the calibration gas, while valve V2 is opened to ambient air in order to prevent overpressure at V1. After calibration, V2 is closed in order to avoid contamination of the calibration gas by ambient air through diffusion and to minimize its consumption.

Within the ion source of the MS, the sample molecules and atoms are ionized by electron impact ionization (70 eV). They are then separated in the quadrupole analyzer based on their mass-to-charge ratio ($m/z$) and ultimately detected using a

Faraday cup. Alternatively, a secondary electron multiplier (SEM) can be used for detection. However, systematic measurements (see below) have shown that for $N_2$ and $O_2$ the standard deviation is half as large and the deviation from the target values is smaller using the Faraday cup (Table 1).



For the quantification of $N_2$, $O_2$ and Ar, the respective ion currents can be extracted from the peaks of the nominal m/z ratios in the mass spectra ($N_2$: $m/z = 28$, $O_2$: $m/z = 32$, Ar: $m/z = 40$). During a 1-s measurement cycle, an ion current is detected for each gas species, which requires a measurement time of 340 ms per $m/z$ ratio.


The calibration with standard gas ($x(N_2)$: 78.1 %, $x(O_2)$: 20.9 %, $x(Ar)$: 1.0 %), is used to calculate the mole fraction ($x$) of each gas species, based on the ion currents and the establishment of calibration factors using Ar as the internal standard. A detailed description of this method is provided in the Appendix A. In this study, the standard gas was calibrated using MS and dry ambient air.

The performance of the GAM 2000-MS was evaluated in 60 repeated measurements of ambient air over a period of 6 h, with ambient air calibration of the system between measurements conducted over the entire period. The average of the 60 measurements was used to calculate the performance parameters of the MS (Table 1).

All valves (V1–V3) were computer controlled.

**Table 1: Performance parameters of the GAM 2000-MS obtained by repeated measurements of the gases in ambient air. $x_d$ is the mole fraction deviation between the measured and the target value. *aSD*: absolute standard deviation, SEM: secondary electron multiplier.**

| Detector | Gas | $x_d$ [$\times 10^{-3}$ %] | *aSD* [$\times 10^{-3}$ %] |
|---|---|---|---|
| Faraday cup | $^{28}N_2$ | 3.68 | 5.49 |
| | $^{32}O_2$ | 3.44 | 5.31 |
| | $^{40}Ar$ | 0.25 | 0.34 |
| SEM | $^{28}N_2$ | 5.58 | 9.44 |
| | $^{32}O_2$ | 5.41 | 9.06 |
| | $^{40}Ar$ | 0.18 | 0.44 |

## 3 Performance of the concentration measurements

### 3.1 Accuracy and precision

High instrument precision and accuracy are critical for measuring biogenic changes in $N_2$ and $O_2$ concentrations. This is particularly important in determinations of $N_2$ fixation in the Baltic Sea, as concentration deficits of only about 5 µmol-$N_2$/L (derived from the depth-integrated $N_2$ fixation in Schneider et al., (2014)) at a background concentration in Baltic seawater of about 500 µmol-$N_2$/L (at 20 °C) can be expected.

The partial pressures of $N_2$, $O_2$ and Ar in the gas room of the membrane equilibrator ($p_i$) can be calculated by a modification

of Dalton's Law (Eq. 1). Since partial mole fractions, which refer only to the total moles of the three considered gases ($x_i'$, see the Appendix A), are used, the total pressure in the headspace ($p_t$) must be corrected for gases that were not measured, such as water vapor. However, the MS-based determination of water vapor is challenging due to the lack of a suitable





calibration medium and the potential for overlapping *m/z* ratios, leading to imprecise measurements. Thus, water vapor is instead determined by assuming saturation at the given temperature and salinity (Ambrose and Lawrenson, 1972). The

contribution of other gases, e.g., $CO_2$, to the total pressure is assumed to be very minor and is neglected. The partial pressure of the considered gases is therefore calculated as shown in Eq. (1):

$$p_i = x_i' \cdot (p_t - pH_2O) \ , \hspace{4cm} (1)$$

Finally, the concentration of dissolved gases ($c_i$) in the water phase of the membrane equilibrator is calculated using the molar gas volume and the Bunsen coefficients, obtained from Hamme and Emerson (2004) for $N_2$ and Ar, and from

Weiss (1970) for $O_2$, as shown in Eq. (2):

$$c_i = \frac{\beta_i \cdot p_i}{V_m} , \hspace{4cm} (2)$$

with:

$\beta$ – Bunsen coefficient

$p$ – partial pressure

$V_m$ – molar gas volume

The accuracy and precision of the concentration measurements were assessed by coupling the GE-MIMS with a temperature-controlled bath (Huber CC-K15) filled with distilled water. The thermostat was set to a constant temperature ($T = 18.5 \pm 0.02°C$) and was open to the atmosphere in order to generate an equilibrium with atmospheric gases. The water

was continuously recirculated through the membrane equilibrator at a flow rate of 2 L/min. Following calibration of the GE-MIMS, the mole fractions of $N_2$, $O_2$ and Ar and the total pressure of the gas phase were continuously measured for 1 h. After an initial adjustment period, the measured values were averaged ($\sim\Delta t = 20–60$ min, Fig. 2) and used to determine the concentration ($c_{meas}$) and standard deviation (*aSD* and *rSD*) of the respective gases (Table 2).

The measured concentrations are presented together with the expected saturation concentrations (Hamme and Emerson,

2004; Weiss, 1970) in Fig. 2 and Table 2. The absolute standard deviation was very similar for $N_2$ and $O_2$, ranging from 0.15 μmol/L to 0.16 μmol/L. The measured concentration of Ar had a a*SD* that was an order of magnitude smaller. Additionally, the data indicated a 0.2 % offset for both $N_2$ and $O_2$ measurements. Due to the lower concentration values, the averaged offset was 0.7 % for Ar, twice as high as that of the other gases (Table 2).



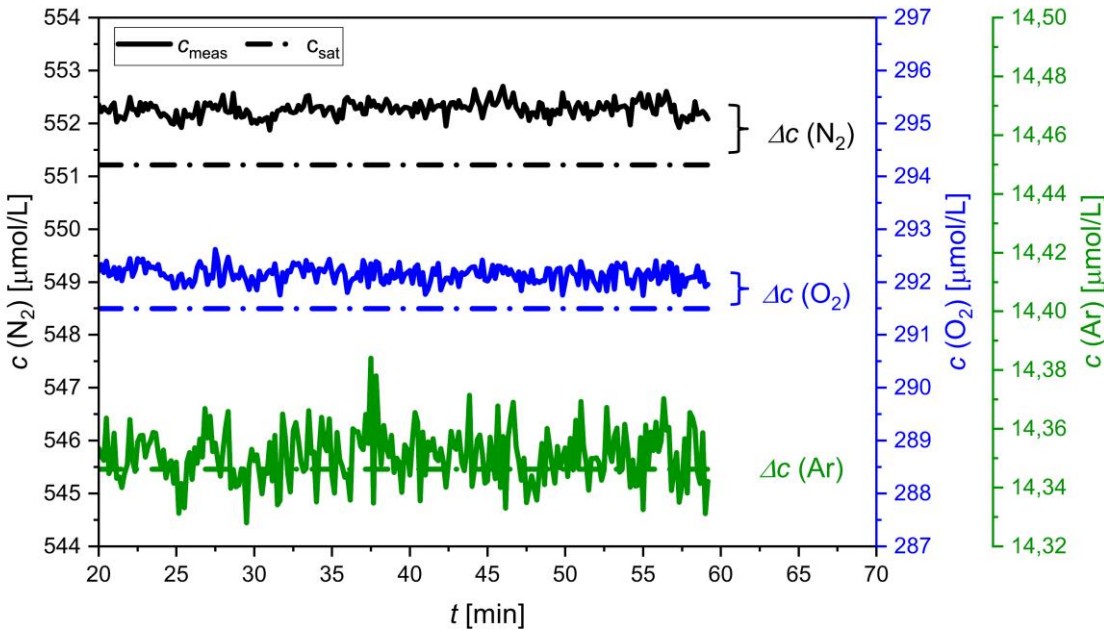

**Figure 2: Concentration measurements of $N_2$ (black), $O_2$ (blue) and Ar (green) in distilled water equilibrated with ambient air during a laboratory experiment. While the measured values ($c_{meas}$) are indicated by solid lines, the saturation values ($c_{sat}$) are represented by the dotted lines.**

The accuracy is given by the concentration difference ($\Delta c$) between the measured value ($c_{meas}$) and the saturation value ($c_{sat}$) of $N_2$, $O_2$ and Ar, as presented in Table 2. The precision is reported as the 2-fold absolute standard deviation ($aSD$).

**Table 2: Results of a laboratory experiment in order to assess the accuracy and precision of the GE-MIMS. $aSD$: absolute standard deviation.**

| Gas | $aSD$ [µmol/L] | $rSD$ [%] | Precision [µmol/L] | Precision [%] | $c_{meas}$ [µmol/L] | $c_{sat}$ [µmol/L] | $\Delta c$ [µmol/L] | $\Delta c$ [%] |
|------|------|------|------|------|------|------|------|------|
| $^{28}N_2$ | 0.15 | 0.03 | 0.30 | 0.05 | 552.3 | 551.2 | 1.1 | 0.2 |
| $^{32}O_2$ | 0.16 | 0.05 | 0.32 | 0.11 | 292.1 | 291.5 | 0.6 | 0.2 |
| $^{40}Ar$ | 0.01 | 0.07 | 0.02 | 0.14 | 14.4 | 14.3 | 0.1 | 0.7 |

Consequently, based on a moderately strong $N_2$ fixation episode of 5 µmol-$N_2$/L (Schneider et al., 2014), $N_2$ fixation can be detected and determined with an accuracy of about 20%. According to the classical Redfield stoichiometry (Redfield et al., 1963) for OM production, the amount of associated oxygen produced through photosynthesis is 66 µmol/L and can be determined with an accuracy of roughly 1 %.



## 3.2 Equilibration kinetics

### 3.2.1 Theory

The following consideration refers to the case in which there is no water flow. A central factor of the equilibration kinetics is the equilibration time $\tau$, which is the time required for equilibrium to be established at a given a transmembrane partial pressure difference, thereby providing information on the temporal resolution of GE-MIMS measurements. The equilibration time can be derived from the general flux equation [Eq. (3)]:

$$\frac{\partial n_g}{\partial t \cdot A} = -k_n \cdot \Delta p \, ,$$  (3)

with:

$\frac{\partial n}{\partial t}$ – change with time of the moles of a gas in the gas side of the equilibrator [mole $\cdot$ s$^{-1}$]

$A$ – membrane area [m$^2$]

$k_n$ – mass (mole) transfer constant [mol $\cdot$ s$^{-1}$ $\cdot$ m$^{-2}$ $\cdot$ atm$^{-1}$]

$\Delta p$ – transmembrane partial pressure difference: $p_g$ - $p_w$ [atm]

subscript g refers to the gas side of the membrane equilibrator and w to the water side

Using the ideal gas law, $\partial n_g$ is replaced by $\partial p_g$ according to Eq. (4):

$$\frac{\partial p_g}{\partial t} = \frac{-k_n \cdot A \cdot R \cdot T}{V_g} \cdot \Delta p \, ,$$  (4)

with:

$\partial p$ - change in the partial pressure of a gas [atm]

$R$ – universal gas constant [m$^3$ $\cdot$ atm $\cdot$ mol$^{-1}$ $\cdot$ K$^{-1}$]

$T$ – absolute Temperature [K]

$V$ –  volume [m$^3$]


To describe $\Delta p$ only as a function of $p_g$, the total moles (gas side + water side) of the considered gas, $n_t$, which is constant in the equilibrator, is introduced, as shown in Eq. (5):

$$n_t = V_g \cdot \frac{p_g}{R \cdot T} + V_w \cdot p_w \cdot s \, ,$$  (5)

with:

$s$ – solubility constant [mol $\cdot$ m$^{-3}$ $\cdot$ atm$^{-1}$] (Hamme and Emerson, 2004)

$p_w$ is thus given as shown in Eq. (6):

$$p_w = \frac{n_t - \left(\frac{p_g}{R \cdot T}\right) \cdot V_g}{V_w \cdot s} \, ,$$  (6)

and $\Delta p$ is expressed using Eq. (7):



$\Delta p = p_{\mathrm{g}} - \dfrac{n_{\mathrm{t}} - \left( \frac{p_{\mathrm{g}}}{R \cdot T} \right) \cdot V_{\mathrm{g}}}{V_{\mathrm{w}} \cdot s}$,  (7)

The differentiation of Eq. (7) yields Eq. (8):

$\partial(\Delta p) = \left( 1 + \dfrac{V_{\mathrm{g}}}{R \cdot T \cdot V_{\mathrm{w}} \cdot s} \right) \partial p_{\mathrm{g}}$,  (8)

Replacing $\partial p_{\mathrm{g}}$ in Eq. (4) then yields Eq. (9):

$\dfrac{\partial(\Delta p)}{\partial t} = -k_n \cdot A \cdot \left( \dfrac{R \cdot T}{V_{\mathrm{g}}} + \dfrac{1}{s \cdot V_{\mathrm{w}}} \right) \cdot \Delta p$,  (9)

The integration of which provides an exponential equation [Eq. (10)]:

$\Delta \mathrm{p} = \Delta p_0 \cdot \exp\left[ -k_n \cdot A \cdot \left( \dfrac{R \cdot T}{V_{\mathrm{g}}} + \dfrac{1}{s \cdot V_{\mathrm{w}}} \right) \cdot t \right]$,  (10)

with a time constant [s$^{-1}$] that equals the reciprocal equilibration time $\dfrac{1}{\tau_{\mathrm{nf}}}$ (no water flow), resulting in Eq. (11) and Eq. (12):

$\Delta \mathrm{p} = \Delta p_0 \cdot \exp\left( \dfrac{-t}{\tau_{\mathrm{nf}}} \right)$,  (11)

$\tau_{\mathrm{nf}} = \dfrac{1}{k_n \cdot A \cdot \left( \frac{R \cdot T}{V_{\mathrm{g}}} + \frac{1}{V_{\mathrm{w}} \cdot s} \right)}$,  (12)

Equation (11) indicates that, after a time $t = \tau_{\mathrm{nf}}$, only $1/e$ of the original difference in the transmembrane partial pressure remains. This implies that the equilibrium state has already reached approximately 63%. Consequently, after four times the equilibration time have elapsed, equilibrium has reached 99%.

In addition to geometric dimensions and thermodynamic properties, gas exchange and thus the equilibration time is controlled by the transfer coefficient $k_n$. According to the membrane's manufacturer, the gas permeability of the

Celgard X50-215 membrane in the Liqui-Cel 1.8 × 8.75 contactor used in our system is 50 Gurley seconds. Gurley seconds are the time that is necessary for 100 cm$^3$ of air to pass through a membrane with an area of 1 inch$^2$ at a pressure difference of 4.88 inches of water. Converting Gurley seconds to common units give a transfer coefficient of $k_n = 11.5$ mol $\cdot$ m$^{-2}$ $\cdot$ s$^{-1}$ $\cdot$ atm$^{-1}$, which refers to air but is considered here to represent the $k_n$ of N$_2$. According to Eq. (12), $k_n$ results in an equilibration time $\tau_{\mathrm{nf}}$ for N$_2$ of $4.6 \cdot 10^{-6}$ s, assuming that no water renewal occurs (no flow). This means the almost-immediate

equilibration of N$_2$ between the dissolved and the gaseous phases. This is plausible given the geometric dimensions of the contactor, which imply that the thicknesses of the water and gas layers ($V_{\mathrm{w}}/A$ and $V_{\mathrm{g}}/A$) are only about 80 µm and 150 µm, respectively, while the membrane is only 40 µm thick.

However, interpretations of the equilibrium must take into account that gas exchange during the equilibration process affects both the gas phase and the dissolved phase. The equilibrium $p_{\mathrm{g}}$ will therefore not be identical with the original $p_{\mathrm{w}}$, as

required by our approach to determine the gas concentration in the water. The increase/decrease in $p_{\mathrm{g}}$ during equilibration in relation to the original $\Delta p$ can be derived as shown in Eq. (13), according to which, at equilibrium, the initial $p_{\mathrm{g}}$ altered by $\partial p_{\mathrm{g}}$ must be equal to the initial $\mathrm{p}_{\mathrm{w}}$ altered by $\partial p_{\mathrm{w}}$ ($\delta p_{\mathrm{w,g}}$ refers to absolute changes):

$p_{\mathrm{g}} - \partial p_{\mathrm{g}} = p_{\mathrm{w}} + \partial p_{\mathrm{w}}$,  (13)

The initial $\Delta p$ can thus be expressed as shown in Eq. (14):



$$\Delta p = \partial p_\mathrm{g} + \partial p_\mathrm{w} \, , \tag{14}$$

$\partial p_\mathrm{g}$ and $\partial p_\mathrm{w}$ can be expressed by the flux of moles across the membrane, as shown in Eq. (15) and Eq. (16):

$$\partial p_\mathrm{g} = \frac{\partial n \cdot R \cdot T}{V_\mathrm{g}} \, , \tag{15}$$

$$\partial p_\mathrm{w} = \frac{\partial n}{V_\mathrm{w} \cdot s} \, , \tag{16}$$

Combining Eq. (15) and Eq. (16) describes $\partial p_\mathrm{w}$ as a function of $\partial p_\mathrm{g}$ and yields Eq. (17):

$$\partial p_\mathrm{w} = \frac{\partial p_\mathrm{g} \cdot V_\mathrm{g}}{V_\mathrm{w} \cdot s \cdot R \cdot T} \, , \tag{17}$$

which together with Eq. (14) yields Eq. (18):

$$\Delta p = \partial p_\mathrm{g} + \frac{\partial p_\mathrm{g} \cdot V_\mathrm{g}}{V_\mathrm{w} \cdot s \cdot R \cdot T} \, , \tag{18}$$

The ratio of $\partial p_\mathrm{g}$ to the original $\Delta p$ is described by Eq. (19):

$$\frac{\partial p_\mathrm{g}}{\Delta p} = \frac{1}{1 + \frac{V_\mathrm{g}}{V_\mathrm{w} \cdot s \cdot R \cdot T}} \, , \tag{19}$$

which shows that, after equilibration, the change in $\partial p_\mathrm{g}$ in relation to the initial $\Delta p$ depends on the ratio of the gas and water volumes and on the solubility of the considered gas. For our Liqui-Cel contactor, the volume ratio is 0.5 and results in a 0.8% change of $\partial p_\mathrm{g}$ with respect to the initial $\Delta p$ whereas, conversely, $\partial p_\mathrm{w}$ changes by 99.2% of the initial $\Delta p$. This means that at equilibrium the partial pressure of the dissolved phase is close to that of the gas phase. However, our approach is based on a gas-phase partial pressure that is at equilibrium with water widely unaffected by gas exchange: $\delta p_\mathrm{g}/\Delta p \approx 1$ or $\delta p_\mathrm{w}/\Delta p \approx 0$. Even a 100-fold increase in the ratio to $V_\mathrm{w}/V_\mathrm{g} = 50$ would only yield a value of 0.45 for $\delta p_\mathrm{g}/\Delta p$. These calculations indicate that the water volume must approach infinity to achieve a 100% adjustment of $p_\mathrm{g}$ to the initial $p_\mathrm{w}$. The latter condition can be approximated by continuously pumping water through the water side of the equilibrator.

A mathematical formulation of the equilibration time for a flow-through system ($\tau$) was derived by considering time steps corresponding to the mean residence time, $t_\mathrm{r}$, of the water in the equilibrator. It is assumed that $t_\mathrm{r}$ is much larger than $\tau$, which is the equilibration time for the no-flow equilibrator [Eq. (12)]. Hence, an approximate equilibrium between the gas and the dissolved phase is generated repeatedly during each renewal of the water in the equilibrator, and the change in $\Delta p$ after each renewal $i$ is identical to the change in $\partial p_\mathrm{g}$ according to Eq. (19). This may be approximated by a differential equation [Eq. (20)] using $i$ as a variable:

$$\frac{\partial (\Delta p)}{\partial i} = \frac{-1}{1 + \frac{V_\mathrm{g}}{V_\mathrm{w} \cdot s \cdot R \cdot T}} \cdot (\Delta p)_i \, , \tag{20}$$

leading to Eq. (21):

$$\partial (\ln \Delta p) = \frac{-\partial i}{1 + \frac{V_\mathrm{g}}{V_\mathrm{w} \cdot s \cdot R \cdot T}} \, , \tag{21}$$

and integrating Eq. (21) yields Eq. (22):



$$\Delta p = \Delta p_0 \cdot \exp\left(\frac{-i}{1+\frac{V_g}{V_w \cdot s \cdot R \cdot T}}\right), \tag{22}$$

Since $i$ is given by the elapsed time, $t$, divided by the residence time, $t_r$, Eq. (22) can be expressed by Eq. (23):


$$\Delta p = \Delta p_0 \cdot \exp\left[\frac{-t}{t_r \cdot \left(1+\frac{V_g}{V_w \cdot s \cdot R \cdot T}\right)}\right], \tag{23}$$

And introducing the residence time at water flow conditions, $\tau$:

$$\Delta p = \Delta p_0 \cdot \exp\left[\frac{-t}{\tau}\right], \tag{24}$$

where $\tau$, is given as shown in Eq. (25):

$$\tau = t_r \cdot \left(1 + \frac{V_g}{V_w \cdot s \cdot R \cdot T}\right), \tag{25}$$

Since the residence time is given by the water volume divided by the water flow rate, $Q_w$:

$$t_r = \frac{V_w}{Q_w}, \tag{26}$$

Eq. (27) is obtained:

$$\tau = \frac{V_w \cdot \left(1+\frac{V_g}{V_w \cdot s \cdot R \cdot T}\right)}{Q_w}, \tag{27}$$

An example is provided as follows: For the Liqui-Cel contactor used in our study and at a flow rate of 2 L/min, $\tau$ is 4.3 min

for $N_2$, 2.2 min for $O_2$ and 2.0 min for Ar. The dependence of the equilibration time on water flow as well as on different

gases (different solubility constants) and $V_g$ values can be seen in Figure 3A, B, which depicts the hyperbolic-shaped

functions described by Eq. (27); a significant increase in $\tau$ occurs at a water flow rate < 500 mL/min. Figure 3A shows that,

due to the solubilities of $O_2$ and Ar, their $\tau$ are very similar, while $\tau$ of $N_2$ is about twice as long. A reduction in the gas

volume of the membrane equilibrator leads to a corresponding decrease in $\tau$ (Figure 3B). The maximum flow of the

membrane used in our study, according to the manufacturer's specifications, is 3000 mL/min. In the following, all

calculations in this section refer to a water temperature of 18 °C and a salinity of 7.



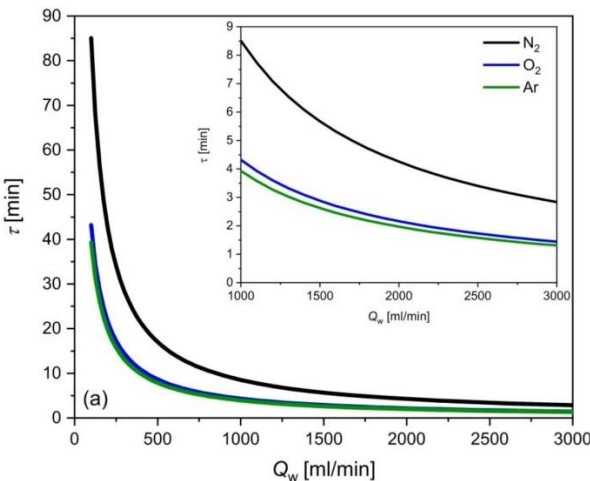
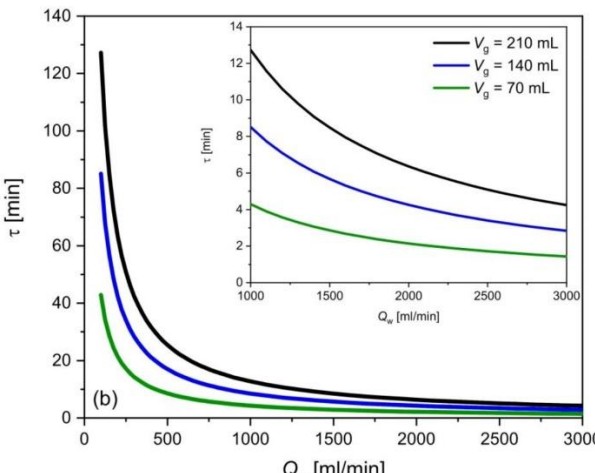

**Figure 3: Theoretically determined dependence of the equilibration time $\tau$ on the water flow rate $Q_w$ using the Liquicel 1.8x8.75 contactor (a) for different gas solubilities ($N_2$, $O_2$ and Ar) and (b) for $N_2$ and different gas side volumes of the membrane equilibrator ($V_g$ of the Liquicel contactor: 140 mL). The insets show the typical flow-rate ranges at higher resolution.**

### 3.2.2 Measurement of $\tau$

The equilibration time ($\tau$) of the membrane equilibrator used in this study (Liqui-Cel $1.7 \times 8.75$) was also experimentally determined at flow-through conditions. For this purpose, 100-L containers were filled with tap water and allowed to rest for at least one day to allow for the development of a homogeneous water mass approximately in equilibrium with the atmosphere. A transmembrane partial pressure difference was generated by initially flushing the gas side of the membrane equilibrator with $N_2$. Subsequently, the water ($T \approx 18°C$) from the container was pumped through the water side at a flow rate of 2 L/min using the peristaltic pump. Adjustments of the $N_2$, $O_2$ and Ar partial pressures in the gas room to match those of these gases dissolved in water were recorded by MS determination of the mole fractions as well as measurement of the total pressure, as described in Sect. 3.1. Figure 5A shows an increase in the $N_2$ partial pressure after the gas side was flushed with $N_2$, followed by a decline due to equilibration with the partial pressure of $N_2$ dissolved in the water. The resulting plateau after approximately 30 min was considered to indicate equilibrium with atmospheric gases. The temporal evolution of the partial pressure difference on the gas side between the value at time $t$, $p_g(t)$, and the value given by the plateaued partial pressure at 30 min, $p_g(\text{eq})$, can be described by the exponential function shown in Eq. (28), which corresponds to Eq. (11):

$$p_g(t) - p_g(\text{eq}) = [p_g(t) - p_g(\text{eq})] \cdot e^{\left(\frac{-t}{\tau}\right)} , \qquad (28)$$

Linearization of Eq. (28) yields an equation where ln $[p_g(t) - p_g(\text{eq})]$ is a linear function of $t$ with a slope that corresponds to the reciprocal equilibration time, Eq. (29):





$\ln[p_{\mathrm{g}}(t) - p_{\mathrm{g}}(\mathrm{eq})] = -\frac{1}{\tau} \cdot t + \ln[p_{\mathrm{g}}(t_0) - p_{\mathrm{g}}(\mathrm{eq})]$ ,                                                                    (29)

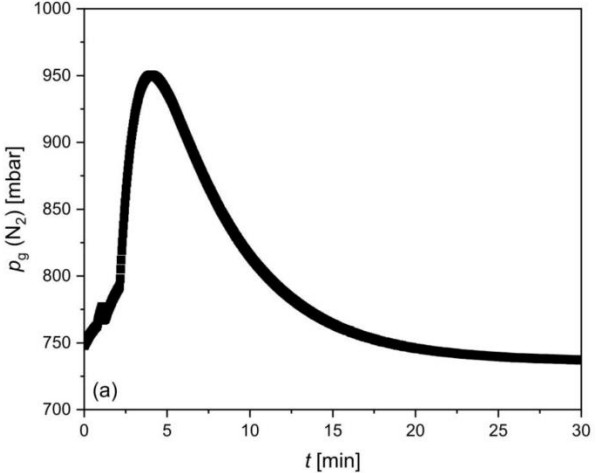
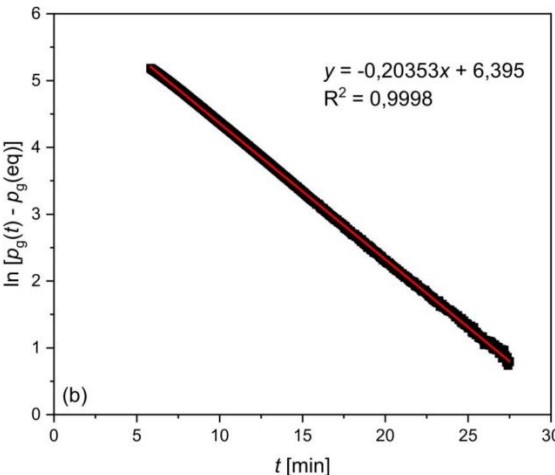

**Figure 4: Results of an experiment to determine the equilibration time for N$_2$. (a) $p_{\mathrm{g}}$ (N$_2$) as a function of time. (b) The logarithmic presentation of the difference between the partial pressure of N$_2$ at any time $t$ and after equilibrium is reached, as a function of**
**time. The reciprocal slope of the regression line represents the $e$-fold equilibration time ($\tau$).**

The equilibration time $\tau$ was then determined from a regression line for $\ln [p_{\mathrm{g}}(t) - p_{\mathrm{g}}(\mathrm{eq})]$ as a function of $t$ (Fig. 5B). The experiment described above was conducted three times, resulting in an average $e$-fold equilibration time ± $aSD$ of $\tau$ (N$_2$) = 4.8 ± 0.1 min, $\tau$ (O$_2$) = 3.2 ± 0.1 min and $\tau$ (Ar) = 3.0 ± 0.2 min. These values are of the same order of magnitude as
those determined theoretically using Eq. (27). The slight deviation is partly due to the assumptions of an ideal gas and an ideal membrane structure without impurities, which does not fully align with the experimental conditions.

The different equilibration times for the considered gases can be attributed to their different solubility constants, according to Eq. (27), with those of O$_2$ and Ar being similar and significantly larger than that of N$_2$. Compared with the $\tau$ (N$_2$/Ar) = 12 min achieved by coupling a MS to a spray-type equilibrator (Schmale et al., 2019), GE-MIMS is nearly three
times faster. Compared with other membrane equilibrator setups, where $\tau$ (Ar) = 4 min (Manning et al., 2016) or $\tau$ (O$_2$/Ar) = 7.75 min (Cassar et al., 2009), our system provides a slightly better temporal resolution.

If the GE-MIMS were installed on a VOS, then the spatial resolution (footprint) would be determined by the equilibration time and the ship's speed. A typical ferry speed is about 20 kn, resulting in a spatial resolution of approximately 12 km for N$_2$ with an equilibration duration of 4 $\tau$. This means that point measurements make no sense and that the obtained gas
concentrations must be averaged over larger sub-areas, since they are subject to permanent ongoing hydrographic dynamics.



## 4 Evaluation of the concentration data

### 4.1 Quantification of biogeochemical effects

Biogeochemically induced changes in $N_2$ or $O_2$ can be estimated using a theoretical reference concentration, i.e., the concentration in the absence of any biogeochemical alterations at the given temperature and salinity. This reference

concentration is not necessarily identical to the saturation concentration, as deviations from the latter can be caused not only by biogeochemical processes (e.g., $N_2$ fixation) but also by physical processes. The latter includes delayed gas exchange at the water surface after temperature- and salinity-dependent changes in solubility, the mixing of different water masses and air bubble injection. However, these effects will not change the ratio of the saturation concentrations of a gas (X) and that of Ar, provided gas X and Ar have identical physicochemical properties (Benson and Parker, 1961; Craig and Hayward, 1987).

Hence, this ratio together with measurements of biogeochemically inert Ar may be used to determine the physically modified saturation concentration of X, as shown in Eq. (30):

$$c_{\text{phy}}(\text{X}) = \frac{c_{\text{sat (X)}}}{c_{\text{sat (Ar)}}} \cdot c_{\text{meas}}(\text{Ar}) \,, \tag{30}$$

where the subscripts phy, sat and meas refer to the physically controlled background, the saturation and the measured concentrations, respectively.


The biogeochemical alteration $\Delta c_{\text{bio}}$ is then obtained according to Eq. (31):

$$\Delta c_{\text{bio}}(\text{X}) = c_{\text{meas}}(\text{X}) - c_{\text{phy}}(\text{X}) \,, \tag{31}$$

which combined with Eq. (30) yields Eq. (32):

$$\Delta c_{\text{bio}}(\text{X}) = \left| \frac{c_{\text{meas (X)}}}{c_{\text{meas (Ar)}}} - \frac{c_{\text{sat (X)}}}{c_{\text{sat (Ar)}}} \right| \cdot c_{\text{meas}}(\text{Ar}) \,, \tag{32}$$

This approach is applicable to $O_2$ and Ar due to their very similar physicochemical properties. Calculation of the biogeochemical changes in the $N_2$ concentration $\Delta c_{\text{bio}}$ ($N_2$) using Eq. (32) has been applied primarily to quantify denitrification in deep waters. However, uncertainties can arise due to the slightly different physicochemical properties of $N_2$ and Ar, mainly with respect to the different temperature dependencies of their solubility constants. As a result, small values of $\Delta c_{\text{bio}}$ ($N_2$), such as those occurring during $N_2$ fixation, can be associated with significant errors (Schmale et al., 2019).

An alternative parameter is the biological saturation parameter $\Delta$X/Ar, which represents the percent deviation from the saturation value caused solely by biogeochemical processes. $\Delta$X/Ar is expressed as shown in Eq. (33):

$$\Delta\text{X/Ar} = \frac{\left(\frac{\text{X}}{\text{Ar}}\right)_{\text{meas}} - \left(\frac{\text{X}}{\text{Ar}}\right)_{\text{sat}}}{\left(\frac{\text{X}}{\text{Ar}}\right)_{\text{sat}}} \cdot 100 \,, \tag{33}$$

For oxygen concentrations both $\Delta c_{\text{bio}}$ ($O_2$) and $\Delta O_2$/Ar were used by Kaiser et al. (2005) and Tortell et al. (2015) to estimate NCP.



### 4.2 Quantification of biogeochemical process rates

The determination of biogeochemical process rates requires measurements of the respective variables as a function of time. The use of a VOS traveling across successive transects as a platform for GE-MIMS would enable the generation of a concentration time series over large areas such as the Baltic Proper (Schneider et al., 2007; Gülzow et al., 2011).

Ignoring vertical mixing across the thermocline, any change in the $N_2$ concentration can be described as the effect of $N_2$ fixation and $N_2$ gas exchange with the atmosphere, using Eq. (34):

$$\Delta N_2 = \Delta N_{2,fix} + \Delta N_{2,gas} \, , \tag{34}$$

The concentration change due to gas exchange ($\Delta N_{2,gas}$) during $\Delta t$ (e.g., the time between two transects) within the mixed-layer depth, $z_{mix}$, is expressed by Eq. (35):

$$\Delta N_{2,gas} = k_{660} \cdot \left(\frac{Sc_{N2}}{660}\right)^{-0.5} \cdot (\bar{N}_{2,sat} - \bar{N}_2) \cdot \frac{\Delta t}{z_{mix}} \, , \tag{35}$$

where $Sc$ is the Schmidt number (Wanninkhof, 1992) and $\bar{N}_{2,sat}$ and $\bar{N}_2$ represent the values of the saturation concentration, averaged over a period of time (e.g., between two consecutive transects) and calculated based on the respective averaged temperature and salinity, and of the measured concentration, respectively. The transfer velocity of gas exchange ($k_{660}$) varies with wind speed and can be estimated using parameterizations such as that suggested by Wanninkhof et al. (2009). However, especially at low wind speeds and during periods of increased OM production, this method is subject to considerable uncertainties. For example, the presence of organic surface films can significantly reduce gas exchange, a phenomenon that current transfer velocity parameterizations do not adequately address. Therefore, concurrent Ar measurements are included to circumvent the uncertainties in using parameterizations of the transfer velocity. This approach is based on the observation that $N_2$ fixation events usually coincide with a significant increase in surface temperature, such that the partial pressure of Ar increases, in turn leading to Ar gas exchange. The change in the Ar concentration due to gas exchange at the water surface can be calculated using Eq. (36), which is similar to Eq. (35):

$$\Delta Ar_{gas} = k_{660} \cdot \left(\frac{Sc_{Ar}}{660}\right)^{-0.5} \cdot (\bar{Ar}_{sat} - \bar{Ar}) \cdot \frac{\Delta t}{z_{mix}} \, , \tag{36}$$

Nitrogen gas exchange can then be quantified without explicitly calculating $k_{660}$, since combining Eq. (35) and (36) yields Eq. (37):

$$\Delta N_{2,gas} = \Delta Ar \cdot \left(\frac{Sc_{N2}}{Sc_{Ar}}\right)^{-0.5} \cdot \frac{\bar{N}_{2,sat} - \bar{N}_2}{\bar{Ar}_{sat} - \bar{Ar}} \, , \tag{37}$$

Finally, the difference in the nitrogen concentration due to $N_2$ fixation for a certain time interval can be obtained using Eq. (38):

$$\Delta N_{2,fix} = \Delta N_2 - \Delta Ar \cdot \left(\frac{Sc_{N2}}{Sc_{Ar}}\right)^{-0.5} \cdot \frac{\bar{N}_{2,sat} - \bar{N}_2}{\bar{Ar}_{sat} - \bar{Ar}} \, , \tag{38}$$

Because of the low wind speed and the surface accumulation of OM during a cyanobacterial bloom, it is assumed that gas exchange is of minor importance and that the temporal change in the measured nitrogen concentration $\Delta N_2$ is the main term (Wasmund, 1997; Lips and Lips, 2008; Schmale et al., 2019).



Eq. (38) includes implicitly the calculation of $k_{660}$ which can be determined explicitly from the Ar measurement by the use of Eq. (36) ($\Delta\text{Ar} = \Delta\text{Ar}_{\text{gas}}$):

$$k_{660} = \frac{\Delta\text{Ar} \cdot z_{\text{mix}}}{(Sc_{\text{Ar}}/660)^{-0.5} \cdot (\overline{\text{Ar}}_{\text{sat}} - \overline{\text{Ar}}) \cdot \Delta t}, \tag{38}$$

Therefore, continuous measurements with our newly developed GE-MIMS system can also be used to determine $k_{660}$,
provided the mixed-layer depth ($z_{\text{mix}}$) can be estimated (Müller et al., 2021).

**5 Conclusion**

The results from our laboratory tests demonstrated that the GE-MIMS system is capable of directly determining cyanobacterial $N_2$ consumption and the associated oxygen production resulting from photosynthesis. Ar concentrations for both the parametrization of air-sea gas exchange and the reconstruction of abiotic background concentrations of $O_2$ and $N_2$
could be measured with high precision and accuracy. Our measurement system is based on the same principle as that of Schmale et al. (2019), but it uses a membrane contactor to generate a true equilibrium between the surface water and air. This results in a significantly shorter equilibration time and thus a higher temporal resolution.

The individual components are designed to allow autonomous operation of the measurement system when installed on a VOS, such as that also currently used for continuous $p\text{CO}_2$ measurements in the Baltic Sea. The resulting $N_2$, $O_2$ and Ar
concentration time series will facilitate determinations of NCP and $N_2$ fixation rates in selected regions of the Baltic Sea. The temporal dynamics of the above-mentioned biogeochemical processes can also be investigated. Furthermore, synchronous measurements of surface $N_2(\text{Ar})$ and $p\text{CO}_2$, as demonstrated herein, take advantage of both the direct determination of $N_2$ consumption by fixation and the high sensitivity of the $\text{CO}_2$ approach to production events. The main limitations of existing approaches to quantifying $N_2$ fixation, which result from the analysis of discrete samples and the use of the elemental
composition of POM, are thus circumvented.

**Appendix A: Evaluation of mass spectrometric data**

In the initial step, ion currents corresponding to the specific mass-to-charge ratios of the following gases are measured using the mass spectrometer: $N_2$; $m/z = 28$, $O_2$; $m/z = 32$, and Ar; $m/z = 40$. We are aware that the $m/z$ ratio for nitrogen may include interferences with other fragment ions, such as from carbon dioxide ($CO^+$). However, given their significantly lower
concentrations, the impact of these interferences is considered negligible.

For calibration a gas mixture characterized by precisely defined molar ratios ($n/n$) of $N_2$, $O_2$ and Ar is used to transform the ion currents ($I$) into mole fractions ($x$). Therefore, calibration factors are determined which are based on the ion currents of $N_2$ and $O_2$ normalized to the ion current of Ar (internal standard). The calibration factors are then obtained by relating the normalized ion currents to the corresponding molar ratio between the considered gas and Ar, as shown in Eq. (39) and
Eq. (40):



$$F_{cal,N2} = \frac{\frac{I_{N2}}{I_{Ar}}}{\frac{n_{N2}}{n_{N2}}} \, , \tag{39}$$

$$F_{cal,O2} = \frac{\frac{I_{O2}}{I_{Ar}}}{\frac{n_{O2}}{n_{Ar}}} \, , \tag{40}$$

Once the calibration factors are determined, measurements of the ion currents for $N_2$, $O_2$ and Ar yield the molar ratios $n_{O2}/n_{Ar}$, $n_{N2}/n_{Ar}$ and, consequently, $n_{O2}/n_{N2}$. These can be used to calculate the mole fractions ($x^{'}$) of $N_2$, $O_2$ and Ar with respect to the sum of $N_2$, $O_2$ and Ar, resulting in Eq. (41) to Eq. (43):

$$\frac{n_{O2}}{n_{Ar}} + \frac{n_{N2}}{n_{Ar}} + 1 = \frac{1}{x'_{Ar}} \, , \tag{41}$$

$$\frac{n_{O2}}{n_{N2}} + \frac{n_{Ar}}{n_{N2}} + 1 = \frac{1}{x'_{N2}} \, , \tag{42}$$

$$\frac{n_{N2}}{n_{O2}} + \frac{n_{Ar}}{n_{O2}} + 1 = \frac{1}{x'_{O2}} \, , \tag{43}$$

Since the analyzed gas, e.g. ambient air, may contain other gases than $N_2$, $O_2$ and Ar, $x^{'}$ is considered as the partial mole fraction. Consequently, $x^{'}$ of $N_2$, $O_2$ and Ar must to be multiplied with the "partial" total pressure which is given by the total pressure ($p_t$) minus the partial pressures of gases such as water vapour, that were not included in the definition of the partial mole fraction, $x^{'}$, e.g. as shown for the partial pressure of $N_2$ in Eq. (44):

$$p_{N2} = x'_{N2} \cdot (p_t - p_{H2O}) \, , \tag{44}$$

## Appendix B: Gas flow into the mass spectrometer vs. gas exchange across the membrane

The permanent gas flow from the membrane equilibrator into the mass spectrometer (MS) will lead to pressure reduction in the gas room and thus potentially prevents the establishment of an equilibrium between the gas phase and the dissolved phase. To estimate the effect of the gas flow on the pressure in the gas room, a steady state is assumed where the removal of air from the gas room, $dn_{out}/dt$, is balanced by the input of dissolved air across the membrane, $dn_{in}/dt$. According to the ideal gas law, $dn_{out}/dt$ can be expressed as shown in Eq. (45):

$$\frac{\partial n_{out}}{\partial t} = \frac{\partial V}{\partial t} \cdot \frac{p_t}{R \cdot T} \, , \tag{45}$$

with:

$p_t$ – total pressure on the gas side

$V$ – volume of the gas side

$T$ – absolute temperature

Since the change in volume over time ($dV/dt$) is equal to the gas flow rate ($Q_g$) into the MS, Eq. (45) can be rearranged to Eq. (46):



$$\frac{\partial n_{\text{out}}}{\partial t} = Q_{\text{g}} \cdot \frac{p_{\text{t}}}{R \cdot T}, \tag{46}$$

If we further assume that the water in the membrane equilibrator is approximately saturated with ambient air, the flux across the membrane, $\partial n_{\text{in}}$, is given by Eq. (47):

$$\frac{\partial n_{\text{in}}}{\partial t} = k_n \cdot A \cdot (p_{\text{atm}} - p_{\text{t}}), \tag{47}$$

with:

$p_{\text{atm}}$ – atmospheric pressure


Equation (46) and Eq. (47) lead to the mass balance described in Eq. (48) for the steady state:

$$Q_{\text{g}} \cdot \frac{p_{\text{t}}}{R \cdot T} = k_{\text{n}} \cdot A \cdot (p_{\text{atm}} - p_{\text{t}}), \tag{48}$$

Finally, an expression can be obtained that describes the deviation of the equilibrium due to the gas flow into the MS, by using the ratio of the total pressure within the gas room ($p_{\text{t}}$) and the pressure of the dissolved phase equilibrated with ambient
air ($p_{\text{atm}}$) as shown in Eq. (49):

$$\frac{p_{\text{t}}}{p_{\text{atm}}} = \frac{1}{1 + \frac{Q_{\text{g}}}{R \cdot T} \cdot k_n \cdot A}, \tag{49}$$

By inserting the parameters of the membrane equilibrator (Liquicel 1.7 x 8.75: $k_n$ = 11.5 mol m$^{-2}$ s$^{-1}$, $A$ = 0.92 m$^2$) and the MS-gas flow ($Q_{\text{g}}$ = 6 µL min$^{-1}$) into Eq. (49), while considering a temperature of 18 °C, a ratio $p_{\text{t}}/p_{\text{atm}}$ = 0.9999 is obtained. Based on this result, it can be reasonably stated that the gas flow into the MS does not significantly affect the equilibrium
within the membrane equilibrator.

**Appendix C: The "2-side equilibration"**

The equilibration of a dissolved gas between the gas phase and the liquid phase occurs by gas exchange which in a closed system affects the partial pressures of the considered gases in both phases. In case of a membrane equilibrator, the gas flux through the membrane of a gas ($F = \frac{\partial n_g}{\partial t \cdot A}$) is proportional to the transmembrane partial pressure difference. The temporal
change of the amounts of gas on the gas side is given by:

$$\frac{\partial n_{\text{g}}}{\partial t} = -k_n \cdot A \cdot \Delta p, \tag{50}$$

with:

$\frac{\partial n_{\text{g}}}{\partial t}$ – change in the amount of gas per time unit on the gas side

$k_n$ – permeability constant

$A$ – membrane area

$\Delta p$ – transmembrane partial pressure difference: $p_g - p_{\text{w}}$





The change in the amount of substance on the gas side is related to the partial pressure by the ideal gas law:

$$\partial n_g = \frac{\partial p_g \cdot V_g}{R \cdot T} \, , \tag{51}$$

with:

$\partial p_g$ – change in partial pressure on the gas side

$V_g$ – volume of the gas side

$R$ – universal gas constant

$T$ – absolute temperature


Combining Eq. (50) and Eq. (51) results in Eq. (52):

$$\frac{\partial p_g}{\partial t} = \frac{-k_n \cdot A \cdot R \cdot T}{V_g} \cdot (p_g - p_w) \, , \tag{52}$$

To express $p_g$ by $p_w$, the total amount of gas ($n_t$) is introduced, as shown in Eq. (53):

$$n_t = c_g \cdot V_g + c_w \cdot V_w \, , \tag{53}$$

with:

$c_g$ – concentration in the gas phase

$c_w$ – concentration in the dissolved phase

$V_w$ – volume of the water side

Whereby $c_g$ and $c_w$ can be calculated as described in Eq. (54) and Eq. (55):

$$c_g = \frac{p_g}{R \cdot T} \, , \tag{54}$$

$$c_w = p_w \cdot s \, , \tag{55}$$

Based on Eq. (53) and using Eq. (54) and Eq. (55), $p_w$ is obtained as shown in Eq. (56):

$$p_w = \frac{n_t - \frac{p_g \cdot V_g}{R \cdot T}}{V_w \cdot s} \, , \tag{56}$$

Inserting Eq. (56) into Eq. (52) finally yields an equation for the change of the partial pressure on the gas side with time:

$$\frac{\partial p_g}{\partial t} = \frac{-k_n \cdot A \cdot R \cdot T \cdot p_g}{V_g} + \frac{k_n \cdot A \cdot R \cdot T \cdot n_t}{V_g \cdot V_w \cdot s} - \frac{k_n \cdot A \cdot p_g}{V_w \cdot s} \, , \tag{57}$$

Introducing $\alpha$ and $\beta$ as auxiliary variables, as given in Eq. (58) and Eq. (59):

$$\alpha = k_n \cdot A \cdot \left( \frac{R \cdot T}{V_g} + \frac{1}{V_w \cdot s} \right) \, , \tag{58}$$

$$\beta = \frac{k_n \cdot A \cdot R \cdot T}{V_g \cdot V_w \cdot s} \, , \tag{59}$$

yields Eq. (60):

$$\frac{\partial p_g}{\partial t} = -\alpha \cdot p_g + n_t \cdot \beta \, , \tag{60}$$

which is integrated via substitution according to Eq. (61) and Eq. (62):





$$x = -\alpha \cdot p_\text{g} + n_\text{t} \cdot \beta \,, \tag{61}$$

$$\partial x = -\alpha \cdot \partial p_\text{g} \,, \tag{62}$$

resulting in Eq. (63):

$$\frac{\partial (\ln x)}{\partial t} = -\alpha \cdot x \,, \tag{63}$$

The integration of Eq. (63) leads to Eq. (64) and Eq. (65):

$$\ln x(t) - \ln x(t_0) = -\alpha \cdot t \,, \tag{64}$$

$$x(t) = x(t_0) \cdot e^{-\alpha \cdot t} \,, \tag{65}$$

After re-substitution the following expressions given by Eq. (66) and Eq. (67) are obtained:

$$-\alpha \cdot p_\text{g}(t) + n_\text{t} \cdot \beta = (-\alpha \cdot p_\text{g}(t_0) + n_\text{t} \cdot \beta) \cdot e^{-\alpha \cdot t} \,, \tag{66}$$

$$p_\text{g}(t) = \left(p_\text{g}(t_0) + n_\text{t} \cdot \frac{\beta}{\alpha}\right) \cdot e^{-\alpha \cdot t} + n_\text{t} \cdot \frac{\beta}{\alpha} \,, \tag{67}$$

$\alpha$ is the time constant of the exponential function, where the reciprocal value represents the equilibration time as shown in Eq. (68):

$$\tau = \frac{1}{\alpha} \,, \tag{68}$$

In case $t$ is approaching infinity, $p_\text{g}$ represents the equilibrium partial pressure, $p_\text{eq}$, according to Eq. (69) and Eq. (70):

$$n_\text{t} \cdot \frac{\beta}{\alpha} = \frac{n_\text{t}}{V_\text{w} \cdot s + \frac{V_\text{g}}{R \cdot T}} \,, \tag{69}$$

$$p_\text{eq} = \frac{n_\text{t}}{V_\text{w} \cdot s + \frac{V_\text{g}}{R \cdot T}} \,, \tag{70}$$

where $n_\text{t}$ at a fixed $\Delta p$ is given by Eq. (71):

$$n_\text{t} = p_\text{g}(t_0) \cdot \frac{V_\text{g}}{R \cdot T} + (\Delta p - p_\text{g}(t_0)) \cdot V_\text{w} \cdot s \,, \tag{71}$$

Re-arranging Eq. (67) with Eq. (68) and Eq. (70) yields:

$$p_\text{g}(t) = \left(p_\text{g}(t_0) - p_\text{eq}\right) \cdot e^{\frac{-t}{\tau}} + p_\text{eq} \,, \tag{72}$$

For the formulation of the counterpart to $p_\text{g}(t)$ which is $p_\text{w}(t)$, the sign of exponential term has to be changed and $p_\text{g}(t_0)$ must be replaced by $p_\text{w}(t_0)$, according to Eq. (73):

$$p_\text{w}(t) = -\left(p_\text{w}(t_0) - p_\text{eq}\right) \cdot e^{\frac{-t}{\tau}} + p_\text{eq} \,, \tag{73}$$

The effect of the "two-sided" equilibration when using our Liquicel membrane contactor with volumes of 70 mL and 140 mL for the water and gas compartment, respectively, is demonstrated in Fig. C1(a). It refers to water that is slightly undersaturated with regard to $N_2$ and shows that at equilibrium, $p_\text{g}$ has only slightly approached $p_\text{w}$. Hence, the system is far away from a state where the partial pressure in the gas phase represents the initial partial pressure of the sample water which

is the basic assumption when determining gas concentrations on the basis of gas phase equilibration. Even when increasing the water volume by a factor of 100 (Fig. C1(b)), the equilibrium partial is still by 38 % above the initial partial pressure of the water. These calculations imply that an infinitely large volume of water is required to fully equilibrate $p_\text{g}$ to the initial $p_\text{w}$.





This condition can be approximated by maintaining a continuous flow of water through the water side of the equilibrator. An Excel spreadsheet for modelling the equilibration process with user-defined variables is provided in the supplementary material.

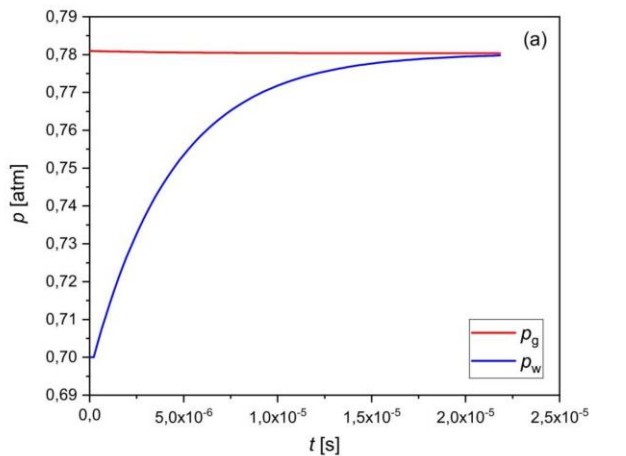 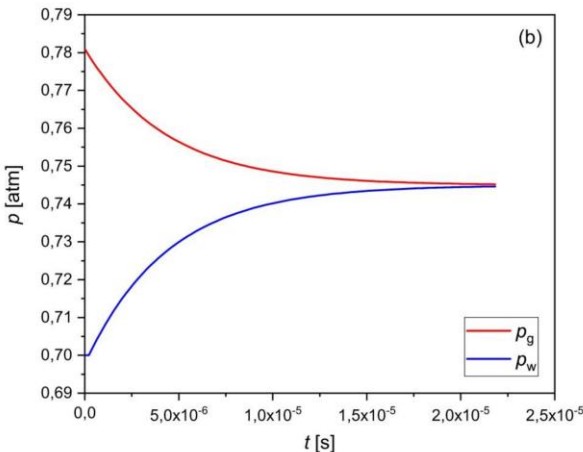

**Figure C1: Changes in the partial pressures on the gas side ($p_g$) and on the water side ($p_w$) over time during an equilibration without water flow using (a) the Liquicel 1.7 x 8.75 membrane contactor. (b) the 100-fold water volume of the Liquicel 1.7 x 8.75 membrane contactor.**

## Data availability

The data used to reproduce the results presented here are archived at http://doi.io-warnemuende.de/10.12754/data-2024-0014 (Iwe, 2024).

## Author contribution

All the authors developed the idea and the design of this manuscript, and SI performed the laboratory experiments. The theoretical considerations of the equilibration process based on the equations were mainly carried out by BS. The paper was mainly written by SI, with major comments and revisions by OS and BS. All the authors contributed to the article and approved the submitted version.

## Competing interests

The authors declare that they have no conflict of interest.



**Acknowledgments**

We thank Bernd Sadkowiak for his assistance in the construction of the measurement setup, Stefan Otto and Michael Glockzin for supporting the laboratory experiments and Sebastian Neubert for programming a data logger (all at IOW).

**Financial support**

This study received funding from the German Research Foundation (SCHM 2530/8-1, SCHN582/9-1).

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
