# Peer review of "Technical note: New approach for the determination of $N_2$ fixation rates by coupling a membrane equilibrator to a mass spectrometer on voluntary observing ships"

_EGUsphere, 2024_

## Referee Comment (RC1)

**Review** of the manuscript *Technical note: New approach for the determination of N2 fixation rates by coupling a membrane equilibrator to a mass spectrometer on voluntary observing ships*, by Sören Iwe, Oliver Schmale and Bernd Schneider, submitted to **EGUsphere**

**Manuscript overview**

The manuscript provides a detailed description of a new method to determine argon (Ar), oxygen ($O_2$) and molecular nitrogen ($N_2$) concentrations in seawater using a membrane contactor to establish gas equilibrium concentrations which are then measured with mass spectrometry. Here argon is used as an inactive compound to calibrate the measurements of $N_2$ and $O_2$. The manuscript also demonstrates increased accuracy of the method compared to previous methods, both in terms of percentage error but also in the time it takes to establish the equilibrium conditions for measurement. Derivation of biological quantities from these measurements are added at the end and the system has not yet been tested in actual operation on a voluntary observation vessel.

**Review overview**

The presented manuscript is very detailed and well written. It is very thorough in its derivation process, set-up and testing description. The method is explicitly presented as suitable for ships of opportunity but this has not been implemented yet. I realise this is presented as a Technical Note, but still I would like to see a bit more context there, i.e. a bit more discussion on actual implementation onboard as well as on the resulting biological quantifications this would allow. I should note that I am not an expert on observational techniques for marine chemistry, and as such, I cannot provide an expert opinion on the presented method though the derivation process seems correct and complete to me.

More detailed comments are provided below.

**Recommendation**

Minor revision

**Detailed Comments**

1. Lines 7-9: here in the abstract some context is mentioned with respect to the importance of N2 fixation as a source for biological activity. Yet this statement is not repeated in the text and references for the assertion are missing. In my opinion, this provides a good context for the presented work and should merit a paragraph in the Introduction, elaborating on the statements and providing references. Now the values are given in line 61 but no context. What are the numbers for riverine N discharge? What for atmospheric deposition of N? And if these are of the same order of magnitude, can we expect a spatial differences in riverine nutrients dominating coastal waters and N2 fixation being a more dominant source offshore? In any case, the abstract cannot contain statements that the manuscript does not substantiate.

2. Line 60: I agree, but even with a larger number of voluntary observational vessels a spatial extrapolation will still be necessary. Using ferry routes is a good start to address the temporal data scarcity, much more than the spatial scarcity.

3. Line 85: missing subscripts in $N_2$ and $O_2$.

4. Line 111: This is the first mention of an appendix, so should be A and not B. Appendix A is only mentioned on line 148.

5. Line 188: there is no explanation of what aSD and rSD actually are. I can guess it, but it should be explicitly mentioned in the text.

6. Table 2: here aSD is explained but rSD still is not, even though it appears in the table.

7. Line 205: the presented accuracy for determining the $N_2$ concentration is high at 0.2% for the used concentration, but the much smaller value representing a "*moderate-strong $N_2$ fixation episode*" generates a related accuracy of 20%. Yet the method is presented as a way to do exactly that: measure $N_2$ fixation to derive biological production based on $N_2$ fixation. Given the derivations in Section 4, how do the authors see this 20% accuracy impacting the ability of the method to quantify the role of $N_2$ fixation in biological N drawdown?

8. Line 316: as the method is specifically aimed at voluntary observational ships, what is the expected impact of varying marine temperature and salinity levels? That is, what part of the technique is sensitive to T, S changes (e.g. solubility constants) and what would that mean for application in other areas? I would prefer to see this discussed in a separate section aimed more explicitly at marine application on ships of opportunity.

9. Line 359-360: the averaging needed over larger spatial scales due to the measurement technique make it suitable for comparison with process-based model results, with usually have a spatial resolution of several km. Point measurements are much less suitable for this. It can also be used to estimate the representativeness of point measurements taken in the vicinity of the transect.

10. Line 386: if 2 articles both used both methods, what are the results from that work? Is one better than the other, or do they differ in accuracy under different circumstances? Now the 2 methods for estimating the biological activity through $O_2$ are mentioned only, leaving the reading guessing what the included references found.

11. Line 394: any $N_2$ input to the surface mixed layer across the thermocline is ignored. Can the authors provide any references for this claim? $N_2$ production through denitrification can occur at depth in low oxygen zones and in sediments. The Baltic is known for the occurrence of extensive "dead zones" due to the limited circulation in the deep basins and the limited exchange with the North Sea. So I would expect $N_2$ production to occur there.

12. Line 408: can the authors provide a reference or explanation for the statement that $N_2$ fixation coincides with a significant increase in surface temperature leading to Ar gas exchange?

13. Line 425: as the aim is to apply this technique on voluntary observational ships, how do the authors propose to estimate the mixed layer depth? Will that be done in situ or afterwards using model results or earth observation tools?

14. Section 4: the authors provide two quantifications using $O_2$ of a proxy for net community production and one estimate for $N_2$ fixation rate (which is stated to be virtually equal to the measured change in $N_2$). It may be outside of the scope of this Technical Note, but it would be good to see some real life testing here using controlled set-ups that allow for an independent quantification of primary production. In the very least this should be proposed as a next step, and could be included in more text about the actual application of the proposed technique onboard. Now these derivations are simply presented as stand-alone results, rather than being tied to the stated objectives and actual implementation of onboard, continuous measurements. Which method of quantification of biogeochemical effects would they recommend for their proposed application? How accurate is the method if first biogeochemical processes (used as a proxy for biological activity) are quantified and then the N2 fixation rate is determined quantifying the role of $N_2$ fixers within the N drawdown associated with primary production?

15. Line 493: again, how do different temperatures affect the equilibrator? 18 $^{\circ}$C seems quite warm for the Baltic and will not represent normal water temperatures entering the water chamber.

---

## Referee Comment (RC2)

**Review of EGUSPHERE-2024-2049 Manuscript**

**Overview and general comments**

The manuscript "New approach for the determination of $N_2$ fixation rates by coupling a membrane equilibrator to a mass spectrometer on voluntary observing ships" describes (i) the design and performance of a GE-MIMS instrument for dissolved gas analysis in surface waters, and (ii) the scientific interpretation of the gas data in terms of the $N_2$ biogeochemistry.

The novelty of the work is not well presented. Much of the manuscript is concerned with replicating in-depth descriptions of previously published work, sometimes without providing credit to these publications. In particular, much of the recent work that developed the GE-MIMS technique is not cited and discussed in the manuscript (for example Patent EP 4 109 092 A1 [1] and other references listed in the detailed comments and at the end of this document). Previously published work should be discussed adequately, and new work done by the authors must be presented to build or expand on these previous work. This will help the authors present the true novelty and relevance of their work (i.e., how they implemented routine analysis of dissolved $N_2$, $O_2$ and Ar in the Baltic Sea with the aim to reduce the uncertainties of previous methods to study the biogeochemical $N_2$ turnover). It should also be mentioned that their experimental work will not only be relevant for the Baltic Sea or for use on "voluntary" ships, and I'd suggest discussing their developments for applications in other oceanic systems, lakes, groundwaters, etc.

I recommend to shorten the manuscript (a lot). I don't see the value of the in-depth (and excessive?) mathematical-theoretical treatise of the assumed gas exchange dynamics in the membrane equilibrator. It seems this treatise is based on inapplicable assumptions, and the modeled equilibration times are inconsistent with the experimental observations. The experimental tests provide all the necessary data without any dependence on the modeling exercise. Also, as the focus of the manuscript lies on the analytical techniques for dissolved gas analysis, the discussion of the theoretical concepts to disentangle the $N_2$ fixation from other processes in the Baltic Sea surface water (Chapter 4) seems out of place. This chapter could be removed and presented elsewhere.

Overall, I can't recommend publication of the manuscript in its current form. The detailed comments below will hopefully prove useful for the authors to revise and improve the manuscript.

**Details and specific comments**

**Title**

I feel the title could be improved to better describe the scope of the manuscript:

- The method is targeted at the analysis of dissolved $N_2$, $O_2$ and Ar in (surface) waters, but this aspect is missing in the title

- Coupling a membrane equilibrator to a mass spectrometer allows dissolved gas analysis, but no direct quantification of $N_2$ fixation rates.

- The techniques described in the manuscript are by no means limited to use on (voluntary) ships

**1. Introduction**

The authors claim (on line 74ff) that their manuscript "introduces the GE-MIMS technique as an extension to MIMS". This is a rather puzzling statement given the extensive previous work that relies on the gas/water equilibrium in a membrane equilibrator. Some of this work is referenced in the manuscript (Cassar et al. 2009, Mächler et al. 2012, Manning et al. 2016). The methods presented in the Cassar and Manning papers allow analysis of the *ratios* of the partial pressures (or concentrations) of different gas species dissolved in the water. The Mächler 2012 work (who introduced the GE-MIMS term) was a first attempt at a semi-quantitative analysis of the absolute partial pressures (or concentrations), which relied on an empirical correction of the analytical data. The GE-MIMS technique was further developed as described in references [4, 5] and Patent EP 4 109 092 A1. This and other potentially relevant works [3, 7, 9] that established the GE-MIMS technique have been ignored in the manuscript.

Line 67 The dynamic steady state in a conventional MIMS is controlled by many more factors than just the dissolved gas concentrations and the MS pumping rate. The water flow rate, the geometry of the membrane system, water salinity, temperature, aging of the membrane material and its gas permeation properties, etc. play a crucial role.

Line 77 Pressure can approach zero (in a vacuum system), but I don't understand how pressure can be negative ("beyond vacuum").

**2.1 Membrane equilibrator**

Figure 1 The gas inlet from the calibration gas tank does not seem to have a pressure controller. However, the gas pressure at the gas inlet to the MS capillary must be

known accurately and precisely to allow reliable calibration of the MS data. How did they achieve this without knowing the pressure of the calibration gas?

Appendix A, line 117/118  Using a pressure sensor to determine the total gas pressure and to quantify the partial pressures of the different gas species in the membrane equilibrator has been previously described in patent EP 4 109 092 A1, which should be referenced here.

Line 107  Which filter? Filter for what, where?

Line 112  How "negligible" is the gas removal? This is a crucial control for the accuracy of the analytical results and calls for a quantitative argument.

Line 114 and 115  Why would a clogged capillary pose a risk for the MS? I'd rather argue that the clogging protects the MS from accidents with too much water.

Line 116 and 117  Is this a confusion between accuracy and precision?

Line 121  Pressure can approach zero (in a vacuum system), but I don't understand how pressure can be negative ("beyond vacuum").

Line 121 and 122  Why would the depressurization in the outflow tubing have an effect on the gas/water equilibrium in the membrane module? Please explain.

**2.2 Mass spectrometry**

Line 128  How important is gas leakage across the walls of the fused silica capillary (transfer of gases from ambient air into the low-pressure internal gas flow of the capillary)?

Line 128  Internal or external diameter?

Line 139/140  The Faraday cup and SEM are likely used not only for detection, but rather for quantification.

Line 140–142  One might expect a better signal/noise ratio from the SEM, in contrast to the observation reported here. Why is this? Please elaborate.

Line 143/144  Quantification of the partial pressures must be based on the peak heights in the mass spectrum. To determine the peak heights, the baseline values therefore need to be subtracted from the peak-top values measured at the indicated m/z positions. Were the baseline values measured? At which m/z values?

Line 143ff  Quantification of the partial pressures cannot be done accurately from the peak heights because their dependence on the total gas pressure at the capillary inlet follows a complicated, non-linear function [6]. With the exception of the special case where the total gas pressures of the sample gas and the calibration gas are identical, the peak-height comparison as described here will therefore not yield accurate results.

Line 145 Why use the same measurement time for all species? Compared to $N_2$ and $O_2$, the much lower abundance of Ar results in a much smaller Ar peak intensity. It therefore seems advisable to use considerably longer measurement times for Ar to optimize the signal/noise ratio.

Line 146 Why not use ambient air as a reference gas for routine calibration? The intermediate step of using a dedicated gas mixture that is cross-calibrated to air seems like an unnecessary step that complicates the analytical setup and potentially introduces additional uncertainty to the data calibration.

Lines 150–154 Why 60 repetitions for averaging? Why a 6 h long test period? The usual approach is to optimize the signal/noise ratio while minimizing the effect of drift. This is commonly done using an Allan plot. Is this what the authors did? Please explain.

Appendix A, line 445 I am not convinced that the CO interference on $m/z = 28$ is negligible for the $N_2$ quantification, especially since $CO_2$ levels in the water may be elevated. Please quantify the potential effect of the CO interference for $N_2$ quantification.

**3.1 Accuracy and Precision**

Line 168/169 Estimating the water vapor pressure by assuming saturation in the GE-MIMS equilibrator has been described in patent EP 4 109 092 A1, which should be referenced here.

Lines 173–180 Using Henry's Law to convert the partial pressures to dissolved gas concentrations has been described in previous GE-MIMS work, which should be referenced here (see previous comments).

Lines 182–188 Air-equilibrated water (AEW) is a good reference to assess the analytical performance, but fabrication of AEW is notoriously difficult. I would recommend comparison and validation of their GE-MIMS system with other (validated and established) methods for dissolved-gas quantification.

192/193 The RSD is normalized relative to the concentration value. A lower concentration value should therefore not result in a lower RSD.

Line 200, Tab. 2 Is the precision reported as the absolute standard deviation (aSD, as indicated in the Table caption) or as the 2-fold of the aSD (as indicated in the text)?

Lines 205–208 I don't understand. What are these numbers? Where do they come from?

**3.2.1 Theory of equilibration kinetics**

I am not convinced that this section adds much value to the manuscript. On the one hand, it assumes that the water is stagnant inside the membrane module (it is not),

and it assumes that the membrane provides the bottleneck for the gas transfer between the water the gas phase. However, the resistance of the membrane material to the gas transfer is marginal (the authors can convince themselves about this by blowing into the water inlet of a dry module while blocking the water outlet, and observe how the air easily escapes through the membrane material into the gas headspace). In contrast, the main bottleneck for the transfer of gas species between the water and the gas headspace is expected to result from the gas exchange mechanisms at the gas/water interface (see for example [8]).

The main outcome of section 3.2.1 is that the partial-pressure equilibration follows an exponential function, which comes to no surprise given the assumption of a first-order exchange kinetic, and which does not warrant any mathematical derivation. A second result is equation (27), which provides a formula to calculate the equilibration time. However, this equation relies on incorrect model assumptions (stagnant water, membrane as bottleneck for gas/water transfer) and therefore does not provide much insight.

Line 233   What is the "solubility constant $s$"? Could this be rewritten to use the Bunsen coefficient $\beta$ introduced before?

Equation (12)   The equilibration time $\tau$ must be a function of the transfer rate $k_n$, which, however, is not shown in equation (12). Please explain.

Equations (13) and (14)   This use of the $\partial p_{g,w}$ notation is rather awkward. By convention, the $\partial$ symbol is used as pairs in fractions to denote partial derivatives. They are not meaningful as isolated elements as used here. The $\partial$ symbols should be replaced by proper differentials ($dp_{g,w}$). This may apply to most other equations, too.

Line 281   The internal diameter of the X50 membrane fibers used in the 3M/Membrana membrane module is $240\,\mu$m [2]. Therefore, the water volume will be smaller than the gas volume by orders of magnitude, not just by a factor of 2.

Lines 285–286   This seems like a trivial finding since the removal of gas from a finite, stagnant volume of water will result in a lower dissolved-gas concentration, and hence in a lower partial pressure at equilibrium. In reality, there's a continuous flow of water through the membrane module, which means there's a (virtually) infinite amount of water available for equilibration with the gas headspace. Again, this shows that the model concept and equations are based on inappropriate assumptions.

**3.2.2 Measurement of $\tau$**

This section provides robust information on the time needed to attain gas/water equilibrium in the membrane module, and provides a useful basis to estimate the spatial resolution of the dissolved-gas data recorded on a moving ship. The measured equilibration times $\tau$ are approximately 50 % higher than those calculated from the model

in Sec. 3.2.1, which supports my impression that the model is inaccurate and seems inappropriate to optimize the operation of the GE-MIMS method for the dissolved gas monitoring described in the manuscript. To this end, the experimentally determined $\tau$ values are more suitable, and the model could be removed from the manuscript entirely.

Lines 324–333 This experimental setup certainly works, but I don't understand why the dissolved-gas concentrations in the water were maintained at a fixed value and the disequilibrium was imposed by changing the partial pressures in the headspace. This approach is backwards to how the GE-MIMS concept works: variations in the aqueous concentrations result in a change of the partial pressures in the headspace. It would seem natural to design the test such that the aqueous concentrations are variable and the response of the partial pressures is monitored to determine the GE-MIMS equilibration time (see for example [3]). Why did the authors choose the "backwards" approach?

Lines 340 I don't see the need for 29 equations simply to state that the partial pressures evolve exponentially towards their equilibrium value. This seems like a trivial result of the assumed first-order gas-exchange kinetic.

Fig. 4 The right panel seems unnecessary, as it shows the same data as the one on the left. I'd suggest to show only the left panel and add the fitted exponential curve.

Lines 349–351 The ratios of the measured and modeled $\tau$ values are $4.8/4.3 = 1.1$ ($N_2$), $3.2/2.2 = 1.5$ ($O_2$), and $3.0/2.0 = 1.5$ (Ar). In other words, the true (measured) values are up to $50\%$ higher than those estimated from the model. I don't see how this large discrepancy can be explained by non-ideality of the gas or "impurities" of the membrane. As mentioned before, there are more fundamental flaws in model assumptions.

Lines 357–360 The membrane module used in this work is rather large and therefore exhibits long equilibration times of 12–20 min. Why did the authors not use much smaller membrane modules that would allow equilibration within about 3 min [4], which would in turn also provide approximately 5× better spatial resolution in their dissolved-gas monitoring?

**4 Evaluation of concentration data**

I feel this chapter is not well integrated in the scope of the otherwise well structured manuscript. Similar to Sec. 3.2.1 it also provides excessive (and seemingly unnecessary) mathematical derivations that seem unnecessary for the purpose of this manuscript. Also, while I am not an expert on $N_2$ geochemistry in surface waters, I would be surprised if these concepts and equations have not been presented and discussed in the

existing literature.

Lines 366–369  I don't agree. The physico-chemical properties of $N_2$ *are* different to those of $O_2$ and Ar, as demonstrated, for example, by the measurements in Sec. 3.2.2. These differences *do* result in fractionation of $N_2/Ar$ relative to $O_2/Ar$.

Section 4.2  The Schmidt-Number method provides a rough estimate of the gas exchange of the mixed layer with the atmosphere. However, if the Schmidt-Number model is really necessary here, I feel this discussion needs to be expanded with a quantitative assessment of the inherent uncertainties.

Lines 418–420  As I understand it, the Schmidt-Number model breaks down at low wind speed, as gas exchange rates do *not* tend to zero without wind. This statement therefore warrants a more quantitative argument based on observed data.

Lines 424–425  This has been demonstrated with a GE-MIMS instrument in previous work [9].

**5. Conclusion**

This chapter will need to be reworked to reflect to focus and new findings of a reworked manuscript.]

**References**

[1] Gas-equilibrium membrane inlet mass spectrometry with accurate quantification of dissolved-gas partial pressures (GE-MIMS-APP). Patent EP 4 109 092 A1, 2022.

[2] S A Ansari, S Chaudhury, P K Mohapatra, S K Aggarwal, and V K Manchanda. Recovery of plutonium from analytical laboratory waste using hollow fiber supported liquid membrane technique. *Separation Science and Technology*, 48(2):208–214, 2012.

[3] Matthias S Brennwald, A P Rinaldi, J Gisiger, A Zappone, and R Kipfer. Gas equilibrium membrane inlet mass spectrometry (GE-MIMS) for water at high pressure. *Geoscientific Instrumentation, Methods and Data Systems*, 13(1), 2024.

[4] Matthias S Brennwald, M Schmidt, J Oser, and R Kipfer. A portable and autonomous mass spectrometric system for on-site environmental gas analysis. *Environmental Science and Technology*, 50(24):13455–13463, 2016.

[5] Eliot Chatton, Thierry Labasque, Jérôme de La Bernardie, Nicolas Guihéneuf, Olivier Bour, and Luc Aquilina. Field continuous measurement of dissolved gases with a CF-MIMS: Applications to the physics and biogeochemistry of groundwater flow. *Environmental Science and Technology*, 51(2):846–854, 2017.

[6] G M Fryer. A theory of gas flow through capillary tubes. *Proc. Royal Soc. London, Ser. A Math. Phys. Sci.*, 293(1434):329–341, 1966.

[7] S Giroud, Y Tomonaga, M S Brennwald, N Takahata, T Shibata, Y Sano, and R Kipfer. New experimental approaches enabling the continuous monitoring of gas species in hydrothermal fluids. *Front. Water*, 4, 2023.

[8] R P Schwarzenbach, P M Gschwend, and D M Imboden. *Air-Water Exchange*. Wiley, 2002.

[9] UW Weber, PG Cook, Matthias S Brennwald, R Kipfer, and TC Stieglitz. A novel approach to quantify air–water gas exchange in shallow surface waters using high-resolution time series of dissolved atmospheric gases. *Environmental Science and Technology*, 53(3):1463–1470, 2018.

---

## Author Response (AR1)

**Response to the Editor:**

Dear Prof. Cook,

please find our revised manuscript "Technical Note: Testing a new approach for the determination of $N_2$ fixation rates by coupling a membrane equilibrator to a mass spectrometer for long term observations" enclosed following major revision. We sincerely thank you and the reviewers for their thoughtful comments that have improved the article. We include our response to the reviews below, and hope that the revised version is suitable for publication in Biogeosciences.

On behalf of our co-authors with best regards,
Sören Iwe

**Response to Reviewer 1 Comments (RC1):**

Description of changes made in the manuscript is denoted with blue.

Review overview

The presented manuscript is very detailed and well written. It is very thorough in its derivation process, set-up and testing description. The method is explicitly presented as suitable for ships of opportunity but this has not been implemented yet. I realise this is presented as a Technical Note, but still I would like to see a bit more context there, i.e. a bit more discussion on actual implementation onboard as well as on the resulting biological quantifications this would allow. I should note that I am not an expert on observational techniques for marine chemistry, and as such, I cannot provide an expert opinion on the presented method though the derivation process seems correct and complete to me. More detailed comments are provided below.

**Reply: We thank the reviewer for the constructive feedback and positive remarks! The manuscript will be revised in order to clarify the biochemical implications of $N_2$ fixation. Furthermore, we will present a more detailed explanation of the practical implementation of the measurement system on a voluntary observing ship (VOS).**

Lines 7-9:

here in the abstract some context is mentioned with respect to the importance of $N_2$ fixation as a source for biological activity. Yet this statement is not repeated in the text and references for the assertion are missing. In my opinion, this provides a good context for the presented work and should merit a paragraph in the Introduction, elaborating on the statements and providing references. Now the values are given in line 61 but no context. What are the numbers for riverine N discharge? What for atmospheric deposition of $N_2$. And if these are of the same order of magnitude, can we expect a spatial differences in riverine nutrients dominating coastal waters and $N_2$ fixation being a more dominant source offshore? In any case, the abstract cannot contain statements that the manuscript does not substantiate.

**Reply: We agree with the reviewer's comments and will make the necessary changes. We will remove the N-budget numbers from the abstract and instead revise the manuscript to ensure better alignment and consistency between the abstract and the main text. Therefore, we will expand the introduction to provide more facts regarding the importance of $N_2$ fixation relative to other nitrogen sources, such as riverine discharge and atmospheric deposition, and include relevant references.**

In the abstract, we removed the second sentence with the N-budget numbers and changed the following sentences accordingly: "However, the estimates of the contribution of $N_2$ fixation to the N budget show a wide range. This is due to interannual variability, significant uncertainties in the various techniques used to determine $N_2$ fixation and in extrapolating local studies to entire basins.".

Line 60:

I agree, but even with a larger number of voluntary observational vessels a spatial extrapolation will still be necessary. Using ferry routes is a good start to address the temporal data scarcity, much more than the spatial scarcity.

**Reply: We agree with the reviewer. Nonetheless, the use of a single VOS represents already considerable progress compared to conventional point measurements in space and time typically conducted from research vessels. The following publications provide an example of such added value: Schneider et al., 2015; Jacobs et al., 2021; Gülzow et al., 2011.**

Line 85:

missing subscripts in N2 and O2.

**Reply: We will correct it in the revised manuscript.**

The subscripts have been added accordingly.

Line 111:

This is the first mention of an appendix, so should be A and not B. Appendix A is only mentioned on line 148.

**Reply: The appendices will be restructured and the order of the references will be adjusted.**

We have revised the note in parentheses in line 111 (old manuscript version) as follows: "(see Appendix A)". Accordingly, the note in line 148 has been updated to: "(calculations are presented in Appendix B).".

Line 188:

there is no explanation of what aSD and rSD actually are. I can guess it, but it should be explicitly mentioned in the text.

**Reply: It will be clarified in the revised manuscript.**

We have revised the sentence as follows to include the missing information about aSD and rSD: "After an initial adjustment period the measured values were averaged ($\Delta t$ ~ 20–60 min, Fig. 2) and used to determine the concentration ($c_{\text{meas}}$), the absolute and the relative standard deviation (*aSD* and *rSD* ) of the concentrations of $N_2$, $O_2$ and Ar (Table 2).".

Table 2:

here aSD is explained but rSD still is not, even though it appears in the table.

**Reply: We will add the missing information in the table caption.**

The table caption now reads: "Results of a laboratory experiment in order to assess the accuracy and precision (2-fold *aSD*) of the GE-MIMS. *aSD*: absolute standard deviation, *rSD*: relative standard deviation.".

Line 205:

the presented accuracy for determining the $N_2$ concentration is high at 0.2% for the used concentration, but the much smaller value representing a "moderate-strong $N_2$ fixation episode" generates a related accuracy of 20%. Yet the method is presented as a way to do exactly that: measure $N_2$ fixation to derive biological production based on $N_2$ fixation. Given the derivations in Section 4, how do the authors see this 20% accuracy impacting the ability of the method to quantify the role of $N_2$ fixation in biological N drawdown?

**Reply: As mentioned by the reviewer the accuracy of the measurement system is considered high. The current accuracy of 20% for moderate-strong $N_2$ fixation episodes is a limitation we must accept, but it reflects the performance of our method. While other methods may not necessarily be more accurate (Wasmund et al., 2005), our approach offers the advantage of higher temporal and spatial resolution. Our main purpose is to measure $N_2$ concentration differences to determine the contribution of $N_2$ fixation to the N budget. The role of the $N_2$ fixation for the total seasonal biological N draw down, including the 20 % uncertainty, will briefly addressed in the revised manuscript.**

During the manuscript revision, we recognized that we would like to demonstrate the role of the $N_2$ fixation for the total seasonal biological N drawdown by relating the NCP triggered by $N_2$ fixation to the total annual N-based NCP. This has been incorporated into the text as follows: "The deviation of the measured $N_2$ concentration ($\Delta c$ ($N_2$) = 1.1 µmol/L, Table 2) from the theoretical saturation values indicates that a moderately strong $N_2$ fixation episode of 5 µmol-$N_2$/L (derived from Schneider et al., 2014a), can be determined with an accuracy of about 20%. This uncertainty refers also to the NCP associated with the $N_2$ fixation which at average conditions contributes by 20 – 26 % to the total annual NCP (Schneider and Müller, 2018).".

Line 316:

as the method is specifically aimed at voluntary observational ships, what is the expected impact of varying marine temperature and salinity levels? That is, what part of the technique is sensitive to T, S changes (e.g. solubility constants) and what would that mean for

application in other areas? I would prefer to see this discussed in a separate section aimed more explicitly at marine application on ships of opportunity.

**Reply: Since many years we are running a fully automated measurement system for the determination of surface water trace gas ($CO_2$, $CH_4$, $N_2O$, $CO$) concentrations (e.g. Schneider et al., 2014b, and references mentioned above). Therefore, our GE-MIMS system will be integrated into an existing infrastructure. Variables that affect the chemical-physical properties of dissolved gases such as $N_2$, $O_2$ and Ar will of course be measured with high accuracy (e.g., temperature, salinity, pressure, see Fig. 1). Still, we will add a short paragraph to the introduction to indicate some of the challenges we are facing when operating our GE-MIMS on a VOS.**

We added the following sentences to the end of section 2.1 to highlight some of the challenges when operating our GE-MIMS on a VOS: "Another aspect to be considered when using GE-MIMS for field studies is the effect of biofouling on membrane properties. Here, we suggest to regularly clean or even replace the membrane to maintain its performance. For field studies where there is a significant temperature difference between the water body under investigation and the laboratory, it is recommended to insulate the equilibrator to prevent the formation of water vapor condensate on the gas side of the membrane.".

Line 359-360:

the averaging needed over larger spatial scales due to the measurement technique make it suitable for comparison with process-based model results, with usually have a spatial resolution of several km. Point measurements are much less suitable for this. It can also be used to estimate the representativeness of point measurements taken in the vicinity of the transect.

**Reply: We agree with the reviewer and will address this in the conclusion of the revised manuscript.**

The following sentence has been added to the end of the conclusion: "Furthermore, the possibility of averaging over larger spatial scales due to the operation of the GE-MIMS on a VOS enhances its compatibility with process-based model results, which typically have a spatial resolution of several kilometers.".

Line 386:

if 2 articles both used both methods, what are the results from that work? Is one better than the other, or do they differ in accuracy under different circumstances? Now the 2 methods for estimating the biological activity through $O_2$ are mentioned only, leaving the reading guessing what the included references found.

**Reply: Since the focus of our manuscript is on the determination of the N budget/fixation, section 4.1 will be deleted. Herewith we are following the recommendation of Reviewer #2.**

We have deleted section 4.1.

Line 394:

any $N_2$ input to the surface mixed layer across the thermocline is ignored. Can the authors provide any references for this claim? $N_2$ production through denitrification can occur at depth in low oxygen zones and in sediments. The Baltic is known for the occurrence of extensive "dead zones" due to the limited circulation in the deep basins and the limited exchange with the North Sea. So I would expect $N_2$ production to occur there.

**Reply: $N_2$ fixation in the Baltic Sea takes place during mid-summer when a shallow surface layer at z < 20 m separates the surface from water below. The development of the cyanobacteria bloom starts at low wind speeds which lead to increasing temperatures up to 22 °C, stabilize the thermocline and suppress mixing with underlying water layers. The underlying water, called intermediate water, may affect the $N_2$ depletion in the surface layer, however, dentrification, oxygen depletion and related phenomena occur below the permanent halocline which prevents mixing with surface water.**

We have revised the sentence and added the following information: "Any change in the $N_2$ concentration can be described as the effect of $N_2$ fixation and $N_2$ gas exchange with the atmosphere (Eq. 20) if vertical mixing across the thermocline is ignored. The latter is justified since $N_2$ fixation typically takes place during low wind speeds (< 5 m/s) which lead to a rising thermocline and warming of the surface layer (up to 22 °C) (Müller et al., 2021).".

Line 408:

can the authors provide a reference or explanation for the statement that $N_2$ fixation coincides with a significant increase in surface temperature leading to Ar gas exchange?

**Reply: See e.g. Schneider et al. (2014) and Schmale et al. (2019) which will be adressed in the revised manuscript.**

We have added the references as mentioned above: "This approach is based on the observation that $N_2$ fixation events usually coincide with a significant increase in surface temperature (Schneider et al., 2014b; Schmale et al., 2019), such that the partial pressure of Ar in the surface water increases, which in turn leads to an Ar flux into the atmosphere.".

Line 425:

as the aim is to apply this technique on voluntary observational ships, how do the authors propose to estimate the mixed layer depth? Will that be done in situ or afterwards using model results or earth observation tools?

**Reply: The estimation of mixed layer depth is based on temperature and salinity modeling (Gräwe et al., 2019), rather than in situ measurements. However, the accuracy of these estimations can be validated using research vessel transects (CTD profiles) or data from Argo floats. We will ensure to make this point clearer in the revised manuscript and reference the relevant literature.**

The sentence has been revised and now reads as follows: "Therefore, continuous measurements with our newly developed GE-MIMS system can also be used to determine $k_{660}$,

provided the mixed-layer depth ($z_{mix}$) can be estimated, e.g., by modelling the surface water temperature and salinity profiles (Gräwe et al., 2019).".

Section 4:

the authors provide two quantifications using $O_2$ of a proxy for net community production and one estimate for $N_2$ fixation rate (which is stated to be virtually equal to the measured change in $N_2$). It may be outside of the scope of this Technical Note, but it would be good to see some real life testing here using controlled set-ups that allow for an independent quantification of primary production. In the very least this should be proposed as a next step, and could be included in more text about the actual application of the proposed technique onboard. Now these derivations are simply presented as stand-alone results, rather than being tied to the stated objectives and actual implementation of onboard, continuous measurements. Which method of quantification of biogeochemical effects would they recommend for their proposed application? How accurate is the method if first biogeochemical processes (used as a proxy for biological activity) are quantified and then the $N_2$ fixation rate is determined quantifying the role of $N_2$ fixers within the N drawdown associated with primary production?

**Reply: As we mentioned above section 4.1 will be deleted to better highlight our novel approach in section 4.2. We appreciate the suggestion and recognize that the effect of $N_2$ fixation can be viewed as a trigger for biological production, which could be compared with other measurement methods, especially with already existing $pCO_2$ measurements onboard of the VOS, where our system could be deployed (e.g., Schneider et al., 2014). Additionally, we argue that combining our Technical Note, focused on a new method for the determination of $N_2$ fixation, with a discussion concerning possible methods for the quantification of primary production would exceed the scope of this paper. Therefore, we have chosen to focus on the testing of the new method in this Technical Note, however, adding plans to use it on a VOS.**

We have deleted section 4.1.

Line 493:

again, how do different temperatures affect the equilibrator? 18 °C seems quite warm for the Baltic and will not represent normal water temperatures entering the water chamber.

**Reply: During the period of $N_2$ fixation, sea surface temperatures are most likely higher than 18 °C (up to 22 °C). While varying temperatures do not affect the system's functionality, they can lead to condensation in the gas room of the equilibrator if the water temperature exceeds the ambient air temperature. In this case, temperature insulation of the membrane equilibrator is required.**

We have added the information about the sea surface temperature in the Baltic Sea to section 4: "The latter is justified since $N_2$ fixation typically takes place during low wind speeds (< 5 m/s) which lead to a rising thermocline and warming of the surface layer (up to 22 °C) (Müller et al., 2021).".

References (Authors)

Gräwe, U., Klingbeil, K., Kelln, J., and Dangendorf, S.: Decomposing Mean Sea Level Rise in a Semi-Enclosed Basin, the Baltic Sea, J. Clim., 32, 3089–3108, https://doi.org/10.1175/JCLI-D-18-0174.1, 2019.

Gülzow, W., Rehder, G., Schneider, B., Deimling, J. S. v., and Sadkowiak, B.: A new method for continuous measurement of methane and carbon dioxide in surface waters using off-axis integrated cavity output spectroscopy (ICOS): An example from the Baltic Sea, Limnol. Oceanogr. Methods, 9, 176–184, https://doi.org/10.4319/lom.2011.9.176, 2011.

Jacobs, E., Bittig, H. C., Gräwe, U., Graves, C. A., Glockzin, M., Müller, J. D., Schneider, B., and Rehder, G.: Upwelling-induced trace gas dynamics in the Baltic Sea inferred from 8 years of autonomous measurements on a ship of opportunity, Biogeosciences, 18, 2679–2709, https://doi.org/10.5194/bg-18-2679-2021, 2021.

Müller, J. D., Schneider, B., Gräwe, U., Fietzek, P., Wallin, M. B., Rutgersson, A., Wasmund, N., Krüger, S., and Rehder, G.: Cyanobacteria net community production in the Baltic Sea as inferred from profiling $p$CO$_2$ measurements, Biogeosciences, 18, 4889–4917, https://doi.org/10.5194/bg-18-4889-2021, 2021.

Schmale, O., Karle, M., Glockzin, M., and Schneider, B.: Potential of Nitrogen/Argon Analysis in Surface Waters in the Examination of Areal Nitrogen Deficits Caused by Nitrogen Fixation, Environ. Sci. Technol., 53, 6869–6876, https://doi.org/10.1021/acs.est.8b06665, 2019.

Schneider, B., Gülzow, W., Sadkowiak, B., and Rehder, G.: Detecting sinks and sources of CO2 and CH4 by ferrybox-based measurements in the Baltic Sea: Three case studies, J. Mar. Syst., 140, 13–25, https://doi.org/10.1016/j.jmarsys.2014.03.014, 2014.

Schneider, B., Buecker, S., Kaitala, S., Maunula, P., and Wasmund, N.: Characteristics of the spring/summer production in the Mecklenburg Bight (Baltic Sea) as revealed by long-term pCO2 data, Oceanologia, 57, 375–385, https://doi.org/10.1016/j.oceano.2015.07.001, 2015.

Schneider, B. and Müller, J. D.: Biogeochemical Transformations in the Baltic Sea: Observations Through Carbon Dioxide Glasses, 1st ed., Springer Oceanography, Springer International Publishing AG, Cham, 110 pp., https://doi.org/10.1007/978-3-319-61699-5, 2018.

Wasmund, N., Nausch, G., Schneider, B., Nagel, K., and Voss, M.: Comparison of nitrogen fixation rates determined with different methods: a study in the Baltic Proper, Mar. Ecol. Prog. Ser., 297, 23–31, https://doi.org/10.3354/meps297023, 2005.

**Response to Reviewer 2 Comments (RC2):**

Overview and general comments:

The manuscript "New approach for the determination of $N_2$ fixation rates by coupling a membrane equilibrator to a mass spectrometer on voluntary observing ships" describes (i) the design and performance of a GE-MIMS instrument for dissolved gas analysis in surface waters, and (ii) the scientific interpretation of the gas data in terms of the $N_2$ biogeochemistry. The novelty of the work is not well presented. Much of the manuscript is concerned with replicating in-depth descriptions of previously published work, sometimes without providing credit to these publications. In particular, much of the recent work that developed the GE-MIMS technique is not cited and discussed in the manuscript (for example Patent EP 4 109 092 A1 [1] and other references listed in the detailed comments and at the end of this document). Previously published work should be discussed adequately, and new work done by the authors must be presented to build or expand on these previous work. This will help the authors present the true novelty and relevance of their work (i.e., how they implemented routine analysis of dissolved $N_2$, $O_2$ and Ar in the Baltic Sea with the aim to reduce the uncertainties of previous methods to study the biogeochemical $N_2$ turnover). It should also be mentioned that their experimental work will not only be relevant for the Baltic Sea or for use on "voluntary" ships, and I'd suggest discussing their developments for applications in other oceanic systems, lakes, groundwaters, etc. I recommend to shorten the manuscript (a lot). I don't see the value of the in-depth (and excessive?) mathematical-theoretical treatise of the assumed gas exchange dynamics in the membrane equilibrator. It seems this treatise is based on inapplicable assumptions, and the modeled equilibration times are inconsistent with the experimental observations. The experimental tests provide all the necessary data without any dependence on the modeling exercise. Also, as the focus of the manuscript lies on the analytical techniques for dissolved gas analysis, the discussion of the theoretical concepts to disentangle the $N_2$ fixation from other processes in the Baltic Sea surface water (Chapter 4) seems out of place. This chapter could be removed and presented elsewhere. Overall, I can't recommend publication of the manuscript in its current form. The detailed comments below will hopefully prove useful for the authors to revise and improve the manuscript.

**Reply: We sincerely appreciate the thoughtful feedback of the reviewer on our manuscript. The comments have provided valuable insights, and we are committed to addressing them thoroughly to improve our manuscript. As suggested, we will expand the existing discussion of earlier GE-MIMS work where appropriate and add further references of scientific publications. We will emphasize the novelty of our work and make it clear that our main goal (besides the pure analytical description) is to present an approach with which $N_2$ fixation can be monitored in higher temporal and spatial resolution (e.g. during long term observations on voluntary observing ships (VOS) in the Baltic Sea). This will include a clearer description of how our approach aims to reduce uncertainties in the determination of $N_2$ fixation rates.**
**Regarding the length of the manuscript, we agree that the mathematical modeling may be excessive and will re-organize the corresponding parts of the manuscript by moving some of the mathematical derivations into the Appendix or to the Supplement.**

**We will also reassess Chapter 4 to reduce its length and to highlight the novelty of our work.**

We have significantly shortened section 3.2.1 and removed section 4.1 entirely. The specific changes are outlined in the following specific comments.

Details and specific comments:

Title

I feel the title could be improved to better describe the scope of the manuscript:

• The method is targeted at the analysis of dissolved $N_2$, $O_2$ and Ar in (surface) waters, but this aspect is missing in the title

• Coupling a membrane equilibrator to a mass spectrometer allows dissolved gas analysis, but no direct quantification of $N_2$ fixation rates.

• The techniques described in the manuscript are by no means limited to use on (voluntary) ships

**Reply: The reviewer is right that the title was too unclear. The new title has been adapted and now reads "Technical note: Testing a new approach for the determination of $N_2$ fixation rates by coupling a membrane equilibrator to a mass spectrometer for long term observations". From this title it is now clear that the focus is on the determination of $N_2$ fixation rates. It is true that we can also determine $O_2$ and Ar concentrations with this method. However, the focus of the new approach explained in this paper is the determination of $N_2$ deficits in surface water and the $N_2$ fixation rates that can be derived from them. That is actually the novelty about the present manuscript. We hope that the rework has succeeded in highlighting this better.**

The title has been changed accordingly and now reads: "Technical note: Testing a new approach for the determination of $N_2$ fixation rates by coupling a membrane equilibrator to a mass spectrometer for long term observations".

1. Introduction

The authors claim (on line 74ff) that their manuscript "introduces the GE-MIMS technique as an extension to MIMS". This is a rather puzzling statement given the extensive previous work that relies on the gas/water equilibrium in a membrane equilibrator. Some of this work is referenced in the manuscript (Cassar et al. 2009, Mächler et al. 2012, Manning et al. 2016). The methods presented in the Cassar and Manning papers allow analysis of the ratios of the partial pressures (or concentrations) of different gas species dissolved in the water. The Mächler 2012 work (who introduced the GE-MIMS term) was a first attempt at a semi-quantitative analysis of the absolute partial pressures (or concentrations), which relied on an empirical correction of the analytical data. The GE-MIMS technique was further developed as

described in references [4, 5] and Patent EP 4 109 092 A1. This and other potentially relevant works [3,7,9] that established the GE-MIMS technique have been ignored in the manuscript.

**Reply: We believe that there has been a misunderstanding, which may have been caused by the imprecise title. Our intention is not to claim that we were the first to use the GE-MIMS technique for determining gas concentrations in water. We acknowledged this by citing relevant studies in the introduction. To avoid further confusion, we will revise the introduction and incorporate the suggested scientific references to clearly acknowledge prior developments in the field.**

We have revised the introduction (from line 74ff, old manuscript version) accordingly: "The present study uses a modification of MIMS, the gas equilibrium-membrane-inlet mass spectrometry (GE-MIMS), which has been developed over the years through extensive work by different research groups. The most significant difference from MIMS is the establishment of a gas-phase equilibrium, which is maintained by the removal of only minor amounts of gas from the gas side of the membrane equilibrator. The mass spectrometric analysis of gases dissolved in water by the use of a membrane equilibrator was first suggested by Cassar et al. (2009) and Manning et al. (2016). Mächler et al. (2012) introduced the term "GE-MIMS" and made a first attempt for a semi-quantitative analysis of equilibrium partial pressures of dissolved gases, which were then related to the concentrations in the dissolved phase through the corresponding solubility constants. Since then, the GE-MIMS technique has been further refined for the quantitative determination of dissolved gas concentrations, as documented in various studies (Brennwald et al., 2016; Chatton et al, 2017, Weber et al., 2018) and Patent EP 4 109 092 A1 (Brennwald and Kipfer, 2022).
Our newly developed measurement system builds upon the established GE-MIMS approach, introducing a different calibration method and adapting it specifically for long term observations (e.g. on VOS) of the surface concentration of $N_2$ in order to detect and quantify $N_2$ fixation.".

Line 67

The dynamic steady state in a conventional MIMS is controlled by many more factors than just the dissolved gas concentrations and the MS pumping rate. The water flow rate, the geometry of the membrane system, water salinity, temperature, aging of the membrane material and its gas permeation properties, etc. play a crucial role.

**Reply: The text will be modified: A steady state in the membrane gas room is generated by the balance between the MS pumping rate (outflow) and the diffusion of the dissolved gas across the membrane (inflow).**

The sentence has been changed accordingly: "As a result, a steady state is generated on the gas side of the equilibrator through the balance between the MS pumping rate (outflow) and the diffusion of the dissolved gases across the membrane (inflow).".

Line 77

Pressure can approach zero (in a vacuum system), but I don't understand how pressure can be negative ("beyond vacuum").

**Reply: The reviewer is right that the sentence is misleading. It now reads: "The latter is maintained by the removal of only minor amounts of gas from the gas space of the membrane."**

The sentence has been restructured and now reads: "The most significant difference from MIMS is the establishment of a gas-phase equilibrium, which is maintained by the removal of only minor amounts of gas from the gas side of the membrane equilibrator.".

2.1 Membrane equilibrator

Figure 1

The gas inlet from the calibration gas tank does not seem to have a pressure controller. However, the gas pressure at the gas inlet to the MS capillary must be known accurately and precisely to allow reliable calibration of the MS data. How did they achieve this without knowing the pressure of the calibration gas?

**Reply: The reviewer has misunderstood the calibration of the MS which is not (!) based on the relationship between the partial pressure of a gas and the respective MS ion current. Therefore, we will describe our calibration procedure in more detail in the revised manuscript:**
**To eliminate smaller temperature or pressure fluctuations within the MS, we use an internal standard (Ar) to determine calibration factors. These are obtained from the ratio $I_X/I_{Ar}$ (ratio of the currents for gas X and Ar) divided by the ratio $n_X/n_{Ar}$ (ratio between the molar amounts of X and Ar in the calibration gas). Calibration factors are hence given by: $F_{cal,X} = (I_X/I_{Ar})/(n_X/n_{Ar})$ where $n_X/n_{Ar}$ are the ratios of the corresponding mole fraction in the calibration gas. From this calibration procedure it follows that elemental ratios X/Ar ($N_2/Ar$, $O_2/Ar$ and $N_2/O_2$) are the primary outcome of our MS measurements.**
**The elemental ratios yield mole fractions for $N_2$, $O_2$ and Ar in the headspace of the membrane equilibrator with respect to the sum of $N_2$, $O_2$ and Ar ("incomplete" or "partial" mole fractions) (calculations are presented in Appendix A). To obtain the partial pressures for $N_2$, $O_2$ and Ar, the "incomplete" mole fractions must be multiplied with the sum of the pressures of the three gases which is given by the total pressure in the head space minus the sum of pressures of other gases. The latter is mainly given by the water vapor and is calculated from water vapor saturation in the gas room of the membrane equilibrator at the temperature and salinity of the water. The effect of other trace gases is ignored due to the minor contributions to the total pressure, e.g. the mean surface water $pCO_2$ is about 400 µatm and thus adds only 0.04 % to the total pressure. The total pressure in the headspace is recorded by a high precision pressure gauge (Fig. 1).**
**Nevertheless, we acknowledge that the ionization process within the MS is inherently pressure-dependent, leading to variations in the ionization ratios of gases under different pressure conditions. To mitigate this, the electron and ion densities in the ion formation region were effectively reduced by adjusting the emission current in the ion source. This adjustment minimizes space charge effects and improves linearity in ion**

**yield and fragmentation across different pressure levels. In fact we will mention in the manuscript that we observed at pressures 200 mbar above the calibration point (atmospheric pressure) the molar fraction of $N_2$ changes relatively by 0.4 %. However, a total equilibrium pressure (total gas tension) of surface seawater of more than 200 mbar above atmospheric pressure, e.g. by biological or temperature effects, can be excluded.**

We have added the following detailed explanation of the calibration process to section 2.2: "We used a standard gas to calibrate the MS regularly (gas composition: $x(N_2)$: 78.1 %, $x(O_2)$: 20.9 %, $x(Ar)$: 1.0 %), which we had previously recalibrated with clean, dry air. Calibration using such a standard gas is particularly important in areas where the standard composition of air is affected by exhaust gases, e.g on a VOS. In environments where air pollution can be ruled out, the ambient air can also be used as the standard (e.g. Cassar et al., 2009; Mächler et al., 2012; Manning et al., 2016).We used Ar as an internal standard in order to reduce the effect of temperature or pressure fluctuations within the MS. The calibration factors are given by the ratios $I_X/I_{Ar}$ (ratio of the currents for gas X and Ar) divided by the ratios $n_x/n_{Ar}$ (ratio between the molar amounts of X and Ar in the standard gas). From this calibration procedure it follows that elemental ratios X/Ar for the analyzed gases ($N_2/Ar$, $O_2/Ar$ and $N_2/O_2$) are the primary outcome of our MS measurements. The elemental ratios then yield mole fractions for $N_2$, $O_2$ and Ar with respect to the sum of $N_2$, $O_2$ and Ar. These mole fractions are called "incomplete" or in case that only water vapor effects the composition of the air "dry" mole fraction (calculations are presented in Appendix B). .
Regarding the performance of the MS, it is important to take into account that the ionization process within the mass spectrometer is inherently pressure-dependent, resulting in variations in the ionization ratios of gases under different pressure conditions. To mitigate this, we effectively reduced the electron and ion density in the ion formation region by adjusting the emission current. This resulted in enhanced linearity in ion yield and fragmentation at different pressures. Indeed, our observations indicate that at pressures 200 mbar above the calibration conditions (atmospheric pressure), the relative change of the molar fraction of $N_2$ was 0.4 %. However, a total equilibrium pressure (total gas tension) of gases dissolved in surface seawater of more than 200 mbar above the atmospheric pressure, e.g., by biological or temperature effects, can be excluded.".

Appendix A, line 117/118

Using a pressure sensor to determine the total gas pressure and to quantify the partial pressures of the different gas species in the membrane equilibrator has been previously described in patent EP 4 109 092 A1, which should be referenced here.

**Reply: As mentioned earlier, we use a different calibration method and a different method to calculate the partial pressures of the analyzed gases compared to the approach described in Patent EP 4 109 092 A1. In our setup, a pressure sensor is used to calculate partial pressures from the ("incomplete") mole fractions of gases in the mixture. This approach is standard practice in equilibrator-based systems (e.g., Schmale et al., 2019; Schneider et al., 2014; Gülzow et al., 2011) and is grounded in well-established physical-chemical principles, which are widely understood and do not require specific referencing.**

Line 107

Which filter? Filter for what, where?

**Reply: The term "filter cartridge" was a mistake in wording and will be corrected to "membrane equilibrator".**

We have made the change as described above.

Line 112

How "negligible" is the gas removal? This is a crucial control for the accuracy of the analytical results and calls for a quantitative argument.

**Reply: The explanation in Appendix B will be modified by using a realistic estimate of the transfer constant, $k_n$, that was derived from the measured equilibration time. On this basis a pressure reduction by 0.8 % was obtained by the continuous removal of gas from the gas room of the equilibrator. This is a minor effect and applies to the calibration and measurements as well.**

As described above, we provide a quantitative argument for the statement in the appendix. It shows that the pressure reduction due to the gas flow into the MS is only 0.2 ‰. The following is stated in Appendix A: "To estimate the effect of the continuous flow of gas into the MS ($6\ \mu L \cdot min^{-1}$) on the pressure in the gas room ($p_g$), the development of a steady state in the gas room is considered. The latter is based on a balance between the gas flow into the MS ($F_{MS}$) and the flux of dissolved gases into the gas room ($F_g$) which are given by Eq. (25) and Eq. (26):

$$F_{MS} = Q_V \cdot \frac{p_g}{R \cdot T}, \; [mol \cdot s^{-1}] \tag{25}$$

$$F_g = k_n \cdot A \cdot (p_{atm} - p_g), \; [mol \cdot s^{-1}] \tag{26}$$

with:
$Q_v$ - volume flow into the MS: $1 \cdot 10^{-10}\ m^3 \cdot s^{-1}$
$k_n$ – transfer coefficient: $2.21 \cdot 10^{-5}\ mol \cdot s^{-1} \cdot m^{-2} \cdot atm^{-1}$ (derived from the experimentally determined equilibration time, see Appendix D)
$A$ – membrane area: $0.92\ m^2$
$R \cdot T = 2.39 \cdot 10^{-2}\ m^3 \cdot atm \cdot mol^{-1}$ ($T = 18\ °C$)
$p_g$ – pressure in the gas room [atm]
$p_{atm}$ – total pressure of the dissolved gases, approximately 1 atm

Equation (25) and Eq. (26) lead to the mass balance described in Eq. (27) for the steady state:

$$Q_V \cdot \frac{p_g}{R \cdot T} = k_n \cdot A \cdot (p_{atm} - p_g), \tag{27}$$

Rearranging Eq. (27) yields an expression that describes the effect of the gas flow into the MS through the ratio between pressure in the gas room ($p_g$) and "true" equilibrium pressure ($p_{atm}$) that was assumed to be 1 atm, as shown in Eq. (28):

$$\frac{p_\text{g}}{p_\text{atm}} = \frac{1}{1+\frac{Q_\text{V}}{R \cdot T \cdot k_\text{n} \cdot A}},$$ (28)

Using the values for the variables in Eq. (28) as given above, results in a ratio $p_\text{g}/p_\text{atm} = 0.9998$ which means that the pressure in the gas room deviated by 0.2 ‰ from the equilibrium total pressure.".

Line 114 and 115

Why would a clogged capillary pose a risk for the MS? I'd rather argue that the clogging protects the MS from accidents with too much water.

**Reply: The reviewer is right, the text is misleading and now reads:**
**"In addition to the Liqui-Cell membrane, we tested a membrane equilibrator from PermSelect (PermSelect 1m$^2$), in which the gas exchange between the water and gas phase is mediated by dense hollow silicon fibres (polydimethylsiloxane, PDMS). Since the gas exchange does not take place across pores, clogging of pores by particles that may hamper the gas flux, is avoided by these membranes. However, our tests with the PermSelect membrane showed that the membrane is unsuitable for our application because, for some reason, water accumulates on the gas side, which could be sucked in through the gas inlet of the MS and thus block the inlet.**

We have changed the text accordingly: "In addition to the Liqui-Cel membrane, we tested a membrane equilibrator produced by PermSelect (PDMSXA-1.0), in which the gas exchange between the water and gas phase is mediated by dense hollow fibers consisting of polydimethylsiloxane (PDMS). Since the gas exchange does not take place across pores, clogging by particles that may hamper the gas flux is avoided by these membranes. However, testing with the PermSelect membrane revealed that it is unsuitable for our application. For reasons that remain unclear, water accumulates on the gas side of the membrane, which could potentially be sucked into the inlet of the MS and block gas flow into it. Furthermore, it affects vacuum stability and interferes with accurate mass spectrometric measurements.".

Line 116 and 117

Is this a confusion between accuracy and precision?

**Reply: The term "accurate" will be removed.**

The sentence now reads: "Water temperature in the GE-MIMS system is measured by a temperature probe (T1, PT100, precision: 0.01°C) located at the inlet of the membrane.".

Line 121

Pressure can approach zero (in a vacuum system), but I don't understand how pressure can be negative ("beyond vacuum").

**Reply: The reviewer is right that the sentence is misleading. It now reads: "This is to prevent gravity from creating a suction effect that reduces the total pressure on the gas side of the membrane and disturbs the gas phase equilibrium."**

The sentence has been changed accordingly: "Tests have shown that otherwise a suction effect occurs on the water side that reduces the total gas pressure in the equilibrator and thus disturbs the gas phase equilibrium.".

Line 121 and 122

Why would the depressurization in the outflow tubing have an effect on the gas/water equilibrium in the membrane module? Please explain.

**Reply: See previous Reply. We need an accurate determination of the gas pressure within the membrane module to calculate concentrations. A suction effect at the water outlet can interfere with this pressure measurement, as we have observed in our tests.**

2.2 Mass spectrometry

Line 128

How important is gas leakage across the walls of the fused silica capillary (transfer of gases from ambient air into the low-pressure internal gas flow of the capillary)?

**Reply: Fused silica capillaries are designed to have very low permeability to larger gas molecules/atoms like $N_2$, $O_2$, Ar, and any potential leakage would be minimal compared to the gas flow within the capillary itself. We have not observed significant deviations in our measurements that would indicate a gas leakage and contamination by ambient air into the system.**

Line 128

Internal or external diameter?

**Reply: We will add "internal" to ensure clarity and accuracy in the description.**

We have made the change as described above.

Line 139/140

The Faraday cup and SEM are likely used not only for detection, but rather for quantification.

**Reply: We will change the sentence to "…ultimately detected and quantified using a Faraday Cup."**

We have revised the sentence as follows: "They are then separated in the quadrupole analyzer based on their mass-to-charge ratio ($m/z$) and ultimately detected and quantified using a Faraday cup.".

Line 140-142

One might expect a better signal/noise ratio from the SEM, in contrast to the observation reported here. Why is this? Please elaborate.

**Reply: The reasons can be manifold, with one possible factor being a greater sensitivity to temperature (Hoffmann et al., 2005; Khan et al., 2018). However, determining the reasons and their specific influence is not the focus of our manuscript. Rather, we refer here to the measurements carried out in the laboratory and their results.**

Line 143/144

Quantification of the partial pressures must be based on the peak heights in the mass spectrum. To determine the peak heights, the baseline values therefore need to be subtracted from the peak-top values measured at the indicated m/z positions. Were the baseline values measured? At which m/z values?

**Reply: Baseline correction was performed using values at m/z = 3. In our laboratory tests, we measured the baseline weekly over a longer period of time and could not detect any significant differences in the signal. However, the stability of the baseline may also depend on the location of the device (e.g. on the VOS) – in this case, we recommend conducting additional tests to take the conditions into account.**

We have added the following sentences to the text: "For our laboratory tests, baseline correction was performed weekly using values at $m/z$ = 3. However, baseline stability may vary depending on the location/platform, where the GE-MIMS is used. We recommend conducting additional tests during field operations to take field specific conditions.".

Line 143ff

Quantification of the partial pressures cannot be done accurately from the peak heights because their dependence on the total gas pressure at the capillary inlet follows a complicated, non-linear function [6]. With the exception of the special case where the total gas pressures of the sample gas and the calibration gas are identical, the peak-height comparison as described here will therefore not yield accurate results.

**Reply: See explanation of our calibration procedure in our reply to the reviewer's comment on Figure 1.**

Line 145

Why use the same measurement time for all species? Compared to $N_2$ and $O_2$, the much lower abundance of Ar results in a much smaller Ar peak intensity. It therefore seems advisable to use considerably longer measurement times for Ar to optimize the signal/noise ratio.

**Reply: The reviewer raises a valid point, and this is certainly something that could be considered in future measurements. However, based on our current setup and as described in the manuscript, we were able to achieve sufficient accuracy and precision for all gases, including Ar, using the same measurement time.**

Line 146

Why not use ambient air as a reference gas for routine calibration? The intermediate step of using a dedicated gas mixture that is cross-calibrated to air seems like an unnecessary step that complicates the analytical setup and potentially introduces additional uncertainty to the data calibration.

**Reply: The reviewer makes a good point. However, on VOS, the composition of ambient air can vary significantly due to factors such as proximity to the engine room and other sources of contamination especially with regard to the O$_2$ content. To avoid potential influences, we chose a cross-calibrated gas mixture for our setup. Nonetheless, we appreciate the suggestion and will clarify in the manuscript that for other deployments with access to stable atmospheric air, the latter can certainly be used for calibration, as demonstrated in studies like Cassar et al. (2009), Mächler et al. (2012), and Manning et al. (2016).**

We have added the following sentences accordingly: "We used a standard gas to calibrate the MS regularly (gas composition: $x$(N$_2$): 78.1 %, $x$(O$_2$): 20.9 %, $x$(Ar): 1.0 %), which we had previously recalibrated with clean, dry air. Calibration using such a standard gas is particularly important in areas where the standard composition of air is affected by exhaust gases, e.g on a VOS. In environments where air pollution can be ruled out, the ambient air can also be used as the standard (e.g. Cassar et al., 2009; Mächler et al., 2012; Manning et al., 2016).".

Line 150-154

Why 60 repetitions for averaging? Why a 6 h long test period? The usual approach is to optimize the signal/noise ratio while minimizing the effect of drift. This is commonly done using an Allan plot. Is this what the authors did? Please explain.

**Reply: The decision to use a 6-hour test period was based on the need to observe potential effects such as temperature variations over a sufficiently long time span. Within this period, no significant drift was observed over one hour, as shown in Figure 2. An Allan plot would indeed better illustrate the stability and drift behavior. We appreciate this suggestion and will include this statistical approach in the revised manuscript.**

The Allan plot in Figure 1R shows the Allan deviation ($\sigma_A$) across 60 pre-averaged measurements (refer to Sec. 2.2) using two detectors (Faraday Cup and SEM) for N$_2$. Each data point represents the variation as more averaged measurements are combined over increasing time intervals ($\tau$, $\tau = 360$ s between two measurements). Although periodic calibration was performed approximately every 30 minutes, the observed minima in the Faraday Cup's curve likely stem from other factors, such as transient stability improvements. Overall, the plot indicates that deviation increases as more measurements are averaged together, with the Faraday Cup showing lower deviations than the SEM due to its reduced baseline noise.
Since the original purpose of creating an Allan plot (to determine an appropriate integration time $\tau$, where noise and drift behavior produce the smallest standard deviations) is not applicable here - given that we are not considering a long-term measurement but rather 60 measurements averaged from 100 data points each, with calibration performed in between - we have decided to not include the plot in the manuscript. All the information that could be derived from the Allan plot are already presented in Table 1, such as the smaller standard deviation when using the Faraday Cup.

[Figure]

**Figure 1R:** Allan deviation plot of $N_2$ measurements using a Faraday Cup (blue) and a Secondary Electron Multiplier (SEM, black) detectors. Each point represents the Allan deviation as a function of averaging time $\tau$, calculated over 60 pre-averaged measurement cycles, each separated by 360 seconds.

Appendix A, line 445

I am not convinced that the CO interference on m/z = 28 is negligible for the $N_2$ quantification, especially since $CO_2$ levels in the water may be elevated. Please quantify the potential effect of the CO interference for $N_2$ quantification.

**Reply: Based on the manuscript of Burlacot et al. (2020), approximately 9.81% of the primary $CO_2$ signal at m/z = 44 is fragmented into the CO ion at m/z = 28. Assuming atmospheric $CO_2$ concentration, this would correspond to around 40 ppm of CO, which could potentially interfere with $N_2$ quantification, given that $N_2$ constitutes 78% of air. However, this level of interference can be considered negligible, as also confirmed by our observations. Additionally, $CO_2$ concentrations in the surface waters of the Baltic Sea are significantly undersaturated during periods of $N_2$ fixation (due to biological production, see Schneider et al. 2007), which would further reduce any potential interference. The references mentioned by the reviewer (Mächler et al., 2012; Brennwald et al., 2016) also did not report significant interference of $N_2$ at m/z = 28 due to $CO_2$ at their study conditions. Nevertheless, we acknowledge that this could be a concern in other study areas with very high $CO_2$ concentrations.**

In the main text, we have added the following sentence to section 2.2: "Interferences with $CO_2$ fragments ($CO^+$, m/z = 28) can be ignored as discussed in Appendix B."
In the new Appendix B we quantify the potential effect of the CO interference accordingly:
"We are aware that the $m/z$ ratio for nitrogen may include interferences with other fragment ions, such as from carbon dioxide ($CO^+$). However, based on the manuscript of Burlacot et al. (2020), only 9.81 % of the primary $CO_2$ signal at $m/z$ = 44 is fragmented into the CO ion at $m/z$ = 28. Assuming $CO_2$ concentration close to atmospheric equilibrium concentrations, this

would correspond to around 40 ppm of CO, which interfere with the $N_2$ quantification (at atmospheric $N_2$ concentrations of 78 %). This level of interference is negligible. Furthermore, regarding envisaged measurements on a VOS in the Baltic Sea, the risk of interference becomes even lower because $CO_2$ in the surface waters of the Baltic Sea is strongly undersaturated with respect to atmospheric $CO_2$ during periods of $N_2$ fixation due to concurrent biological production (Schneider et al., 2007).".

3.1 Accuracy and Precision

Line 168/169

Estimating the water vapor pressure by assuming saturation in the GE-MIMS equilibrator has been described in patent EP 4 109 092 A1, which should be referenced here.

**Reply: This is an obvious assumption and has also been used in many studies concerning the determination of partial pressures of gases by the use of equilibrators (e.g. Schneider et al., 2007; Gülzow et al., 2011; Schmale et al., 2019). It does not need to be referenced.**

Line 173-180

Using Henry's Law to convert the partial pressures to dissolved gas concentrations has been described in previous GE-MIMS work, which should be referenced here (see previous comments).

**Reply: This is common practice in all approaches to derive gas concentrations from a gas phase at equilibrium with a dissolved gas. It is based on basic physical-chemical knowledge and does not need to be referenced.**

Lines 182-188

Air-equilibrated water (AEW) is a good reference to assess the analytical performance, but fabrication of AEW is notoriously difficult. I would recommend comparison and validation of their GE-MIMS system with other (validated and established) methods for dissolved-gas quantification.

**Reply: Through our experimental setup, we ensured the production of air-equilibrated water, as indicated by the stability of the measurement values shown in Figure 2. We have evaluated the system as described, and based on our observations, we consider additional evaluations unnecessary at this stage.**

Lines 192/193

The RSD is normalized relative to the concentration value. A lower concentration value should therefore not result in a lower RSD.

**Reply: We agree with the reviewer. This can also be clearly seen in Table 2.**

Line 200, Tab. 2

Is the precision reported as the absolute standard deviation (*aSD*, as indicated in the Table caption) or as the 2-fold of the *aSD* (as indicated in the text)?

**Reply: We will revise the table caption to explicitly state that the reported precision is based on the 2-fold absolute standard deviation.**

The table caption now reads: "Results of a laboratory experiment in order to assess the accuracy and precision (2-fold *aSD*) of the GE-MIMS. *aSD*: absolute standard deviation, *rSD*: relative standard deviation.".

Lines 205-208

I don't understand. What are these numbers? Where do they come from?

**Reply: We understand that this section is unclear, and we will clarify in the revised manuscript by indicating in parentheses that these numbers are derived from the measured accuracy for the referenced biogeochemical concentration changes.**

We have adjusted the text accordingly: "The deviation of the measured $N_2$ concentration ($\Delta c$ ($N_2$) = 1.1 µmol/L, Table 2) from the theoretical saturation values indicates that a moderately strong $N_2$ fixation episode of 5 µmol-$N_2$/L (derived from Schneider et al., 2014a), can be determined with an accuracy of about 20 %.".

3.2.1 Theory of equilibration kinetics

I am not convinced that this section adds much value to the manuscript. On the one hand, it assumes that the water is stagnant inside the membrane module (it is not), and it assumes that the membrane provides the bottleneck for the gas transfer between the water and the gas phase. However, the resistance of the membrane material to the gas transfer is marginal (the authors can convince themselves about this by blowing into the water inlet of a dry module while blocking the water outlet, and observe how the air easily escapes through the membrane material into the gas headspace). In contrast, the main bottleneck for the transfer of gas species between the water and the gas headspace is expected to result from the gas exchange mechanisms at the gas/water interface (see for example [8]). The main outcome of section 3.2.1 is that the partial-pressure equilibration follows an exponential function, which comes to no surprise given the assumption of a first-order exchange kinetic, and which does not warrant any mathematical derivation. A second result is equation (27), which provides a formula to calculate the equilibration time. However, this equation relies on incorrect model assumptions (stagnant water, membrane as bottleneck for gas/water transfer) and therefore does not provide much insight.

**Reply: We agree with the reviewers comment, that the crucial step for the exchange of gases across the membrane is not the diffusion of the gas along the pores of the membrane. It is rather the transfer of the gas across the water/gas interface that controls the flux. The latter depends on many variables and information for the LiquiCel membranes are not available. We have therefore moved the derivation of the equilibration time for stagnant water (no water flow) in the equilibrator, including the use of the gas-gas permeability derived from the Gurley seconds, to the Appendix. The value of this Appendix section lies in its attempt to provide a clear and accessible**

**explanation for the reader - who may not necessarily be an expert in the field - about how equilibrium is established within a membrane equilibrator and the factors that influence this process. We aim to illustrate the underlying principles of gas transfer and equilibration dynamics, which can enhance the reader's understanding of the system.**
**In view of the geometric dimension of the water and gas layers within the equilibrator (thicknesses of 140 μm and 70 μm, respectively), it seems likely that equilibration between the two phases is established during the residence of the water in the equilibrator. At a water flow rates of 1 L/min - 2 L/min the residence time ranges between 7 s and 14 s. For this case that equilibrium is generated during each water renewal, we have derived a mathematical formulation for the dependency of the equilibration time on the water flow rate. This derivation, given in the main text is considered as a first approximation for the theoretical determination of the equilibration time.**

We have moved the derivation of the equlibration time for stagnant water (no water flow) in the equilibrator to Appendix C. Based on the resulting equation [Eq. (44)] and the experimentally determined equilibration times (see section 3.2.2), we calculated the transfer coefficient, $k_n$. The following is stated in Appendix C: "To calculate the transfer coefficient, $k_n$, we first derive an equation for the equilibration time, $\tau_{nf}$, for the hypothetical case in which there is no water flow (see Sec. 3.2.1). The flux across the membrane is driven by the partial pressure difference according to the general flux equation [Eq. (35)]:

$$\frac{\partial n_g}{\partial t \cdot A} = -k_n \cdot \Delta p ,$$ (35)

with:
$\frac{\partial n}{\partial t}$ – change with time of the moles of a gas in the gas side of the equilibrator [mole $\cdot$ s$^{-1}$]
$A$ – membrane area [m$^2$]
$k_n$ – mass (mole) transfer constant [mol $\cdot$ s$^{-1} \cdot$ m$^{-2} \cdot$ atm$^{-1}$]
$\Delta p$ – partial pressure difference: $p_g$ - $p_w$ [atm]
subscript g refers to the gas side of the membrane equilibrator and w to the water side

Using the ideal gas law, $\partial n_g$ is replaced by $\partial p_g$ according to Eq. (36):
$$\frac{\partial p_g}{\partial t} = \frac{-k_n \cdot A \cdot R \cdot T}{V_g} \cdot \Delta p ,$$ (36)
with:
$\partial p$ - change in the partial pressure of a gas [atm]
$R$ – universal gas constant [m$^3 \cdot$ atm $\cdot$ mol$^{-1} \cdot$ K$^{-1}$]
$T$ – absolute Temperature [K]
$V$ – volume [m$^3$]

To describe $\Delta p$ only as a function of $p_g$, the total moles (gas side + water side) of the considered gas, $n_t$, which is constant at zero flow, is introduced, as shown in Eq. (37):
$$n_t = V_g \cdot \frac{p_g}{R \cdot T} + V_w \cdot p_w \cdot s ,$$ (37)
with:
$s$ – solubility constant [mol $\cdot$ m$^{-3} \cdot$ atm$^{-1}$]

$p_w$ is thus given as shown in Eq. (38):
$$p_w = \frac{n_t - \left(\frac{p_g}{R \cdot T}\right) \cdot V_g}{V_w \cdot s} ,$$ (38)
and $\Delta p$ is expressed using Eq. (39):

$$\Delta p = p_{\text{g}} - \frac{n_{\text{t}} - \left(\frac{p_{\text{g}}}{R \cdot T}\right) \cdot V_{\text{g}}}{V_{\text{w}} \cdot s},$$ (39)

The differentiation of Eq. (39) yields Eq. (40):

$$d(\Delta p) = \left(1 + \frac{V_{\text{g}}}{R \cdot T \cdot V_{\text{w}} \cdot s}\right) dp_{\text{g}},$$ (40)

Replacing $\partial p_{\text{g}}$ in Eq. (36) then yields Eq. (41):

$$\frac{d(\Delta p)}{dt} = -k_n \cdot A \cdot \left(\frac{R \cdot T}{V_{\text{g}}} + \frac{1}{s \cdot V_{\text{w}}}\right) \cdot \Delta p,$$ (41)

The integration of which provides an exponential equation [Eq. (42)]:

$$\Delta p = \Delta p_0 \cdot \exp\left[-k_n \cdot A \cdot \left(\frac{R \cdot T}{V_{\text{g}}} + \frac{1}{s \cdot V_{\text{w}}}\right) \cdot t\right],$$ (42)

with a time constant [s$^{-1}$] that equals the reciprocal equilibration time $\frac{1}{\tau_{\text{nf}}}$ (no water flow), resulting in Eq. (43) and Eq. (44):

$$\Delta p = \Delta p_0 \cdot \exp\left(\frac{-t}{\tau_{\text{nf}}}\right),$$ (43)

$$\tau_{\text{nf}} = \frac{1}{k_n \cdot A \cdot \left(\frac{R \cdot T}{V_{\text{g}}} + \frac{1}{V_{\text{w}} \cdot s}\right)},$$ (44)

In addition to the geometric dimensions ($V_{\text{g}} = 1.40 \cdot 10^{-6}$ m$^3$, $A = 0.92$ m$^2$) of the membrane equilibrator and the thermodynamic properties, the gas exchange and thus the equilibration time is controlled by the transfer coefficient $k_n$. The latter can be calculated, using the experimentally determined equilibration times ($\tau$ (N$_2$) = 288 s, Sec. 3.2.2). Since these were determined with a water flow, we assume that $V_{\text{w}}$ is infinitely large, thereby modifying Eq. (44) to yield Eq. (45) and thus $k_n$ for N$_2$:

$$k_n = \frac{V_{\text{g}}}{\tau \cdot A \cdot R \cdot T},$$ (45)

$$k_n = 2.21 \cdot 10^{-5} \text{ mol} \cdot \text{m}^{-2} \cdot \text{s}^{-1} \cdot \text{atm}^{-1}.$$

Additionally, we have elaborated more thoroughly on the assumptions upon the model is based, first in section 3.2.1: "Therefore, it is assumed that during each time step $t_{\text{r}}$ an approximate equilibrium between the gas and the dissolved phase is repeatedly generated. This is a plausible assumption in view of the geometric dimensions of the membrane equilibrator, which imply that the thickness of the water and gas layers, given by the volumes of the water and gas side divided by the area of the membrane, amount to only 80 μm and 150 μm, respectively."

and then in section 3.2.2: "Critical points are that the model does not consider continuous water flow, but the transport of discrete water parcels through the equilibrator, and the assumption that a perfect equilibrium was generated repeatedly after each renewal (residence time) of the water in the membrane equilibrator.".

Line 233

What is the "solubility constant $s$"? Could this be rewritten to use the Bunsen coefficient $\beta$ introduced before?

**Reply: We will consistently use the solubility constant $s$ given as [mol $\cdot$ L$^{-1}$ $\cdot$ atm$^{-1}$].**

The text has been revised so that only the solubility constant $s$ is used.

Equation (12)

The equilibration time $\tau$ must be a function of the transfer rate $k_n$, which, however, is not shown in equation (12). Please explain.

**Reply: The transfer rate $k_n$ is indeed included in Equation (12).**

Equations (13) and (14)

This use of the $\partial p_{g,w}$ notation is rather awkward. By convention, the $\partial$ symbol is used as pairs in fractions to denote partial derivatives. They are not meaningful as isolated elements as used here. The $\partial$ symbols should be replaced by proper differentials ($dp_{g,w}$). This may apply to most other equations, too.

**Reply: We agree with the reviewer and will make the necessary adjustments in the revised manuscript.**

We have made the adjustments accordingly.

Line 281

The internal diameter of the X50 membrane fibers used in the 3M/Membrana membrane module is 240 μm [2]. Therefore, the water volume will be smaller than the gas volume by orders of magnitude, not just by a factor of 2.

**Reply: For our model, we refer to the manufacturer's data sheet (data sheet: 3M Liquicel MM-1.7x8.75 Series Membrane Contactor, 2021), which specifies a water volume (lumen side) of 70 mL and a gas volume (shell side) of 140 mL. This indicates that the water volume is indeed smaller than the gas volume by a factor of 2. We will reference the datasheet in the revised manuscript.**

We have cited the reference in the sentence as follows: "For our Liqui-Celmembrane, the volume ratio is 0.5 (3M data sheet, 2021) and results in a 0.8% change of $\Delta p_g$ with respect to the initial $\Delta p_0$ whereas, conversely, $\Delta p_w$ changes by 99.2% of the initial $\Delta p_0$.".

Lines 285-286

This seems like a trivial finding since the removal of gas from a finite, stagnant volume of water will result in a lower dissolved-gas concentration, and hence in a lower partial pressure at equilibrium. In reality, there's a continuous flow of water through the membrane module, which means there's a (virtually) infinite amount of water available for equilibration with the gas headspace. Again, this shows that the model concept and equations are based on inappropriate assumptions.

**Reply: We acknowledge the reviewer's concerns regarding the model assumptions (see reply above). The purpose of using a model that examines gas exchange with stagnant water was to provide a simplified representation for understanding the fundamental principles of gas dynamics. This approach was intended to help readers, especially those less familiar with the topic, grasp the essential mechanisms at play. We have moved this model to the appendix to clarify its role and significance in our overall analysis.**

3.2.2 Measurement of $\tau$

This section provides robust information on the time needed to attain gas/water equilibrium in the membrane module, and provides a useful basis to estimate the spatial resolution of the dissolved-gas data recorded on a moving ship. The measured equilibration times $\tau$ are approximately 50 % higher than those calculated from the model 5 in Sec. 3.2.1, which supports my impression that the model is inaccurate and seems inappropriate to optimize the operation of the GE-MIMS method for the dissolved gas monitoring described in the manuscript. To this end, the experimentally determined $\tau$ values are more suitable, and the model could be removed from the manuscript entirely.

**Reply: While we acknowledge that the experimentally determined $\tau$ values are more suitable for precise monitoring, we think that it is an established scientific practice to compare experimental results with the outcome of theoretical consideration even if the latter are based on simplified assumptions. Compared with the measurements, our model results yielded the same order of magnitude for $\tau$ of the three gases. Still, significant discrepancies exist between the measured and modeled $\tau$ values which can be attributed to the simplified assumptions inherent in the model. A critical point in the model is for example the assumption that no continuous water flow exists, but that discrete water parcels are transported through the water side of the equilibrator.**

We have added the following sentences accordingly: "These values differ fromthose determined theoretically using Eq. (17), but show the same order of magnitude. The deviations are attributed to the simplified assumptions inherent in the model. Critical points are that the model does not consider continuous water flow, but the transport of discrete water parcels through the equilibrator, and the assumption that a perfect equilibrium was generated repeatedly after each renewal (residence time) of the water in the membrane equilibrator.".

Lines 324-333

This experimental setup certainly works, but I don't understand why the dissolved gas concentrations in the water were maintained at a fixed value and the disequilibrium was imposed by changing the partial pressures in the headspace. This approach is backwards to how the GE-MIMS concept works: variations in the aqueous concentrations result in a change of the partial pressures in the headspace. It would seem natural to design the test such that the aqueous concentrations are variable and the response of the partial pressures is monitored to determine the GE-MIMS equilibration time (see for example [3]). Why did the authors choose the "backwards" approach?

**Reply: We do not understand why our method is a "backward approach". It is exactly the same procedure that is also used in the GE-MIMS method. At a given partial pressure on the gas side, a partial pressure difference is generated by a water flow with a different partial pressure. The equilibration process is then recorded by the change of the partial pressure in the gas phase. A similar "backward" concept was described in [3] where the gas side was filled with pure helium and the equilibration was followed by the diffusion of a gas dissolve in a flow of water into the He gas phase characterized by the flow of water and the gases dissolved in it.**

Lines 340

I don't see the need for 29 equations simply to state that the partial pressures evolve exponentially towards their equilibrium value. This seems like a trivial result of the assumed first-order gas-exchange kinetic.

**Reply: As mentioned above, we have moved the first part of the equations to the appendix. In addition, we believe that the 29 equations describe more than just the exponential evolution of partial pressures within the membrane equilibrator. Our aim in submitting the paper to Biogeosciences was to reach readers who may be less familiar with the theoretical background. By providing more comprehensive derivations and explanations, we hope to give the reader a better insight into the subject.**

As mentioned above, the equations 3 – 12 were added to Appendix C.

Fig. 4

The right panel seems unnecessary, as it shows the same data as the one on the left. I'd suggest to show only the left panel and add the fitted exponential curve.

**Reply: The two panels do not show the same data, although they are based on the same data set. The right panel presents only a selected segment from the left panel, focusing on a specific part of the experiment rather than the entire duration. The reviewer's suggestion may reduce clarity, so for better visualization and understanding, we prefer to keep both figures as they currently are.**

Lines 349-351

The ratios of the measured and modeled $\tau$ values are 4.8/4.3 = 1.1 (N2), 3.2/2.2 = 1.5 (O2), and 3.0/2.0 = 1.5 (Ar). In other words, the true (measured) values are up to 50 % higher than those estimated from the model. I don't see how this large discrepancy can be explained by non-ideality of the gas or "impurities" of the membrane. As mentioned before, there are more fundamental flaws in model assumptions.

**Reply: We refer to the comments above regarding the simplified model, which will be described more detailed in the revised version.**

We have added the following sentences accordingly: "These values differ fromthose determined theoretically using Eq. (17), but show the same order of magnitude. The deviations are attributed to the simplified assumptions inherent in the model. Critical points are that the model does not consider continuous water flow, but the transport of discrete water parcels through the equilibrator, and the assumption that a perfect equilibrium was generated repeatedly after each renewal (residence time) of the water in the membrane equilibrator.".

Lines 357-360

The membrane module used in this work is rather large and therefore exhibits long equilibration times of 12–20 min. Why did the authors not use much smaller membrane modules that would allow equilibration within about 3 min [4], which would in turn also provide approximately 5x better spatial resolution in their dissolved-gas monitoring?

**Reply: The larger gas volume was intentionally chosen to ensure that the equilibrium is not disturbed, as explained in our response to the reviewer's comment on line 112. We also clarify this in our manuscript in lines 111-112. Furthermore, the equilibration time is not as critical for the intended field studies, since a reasonable data evaluation requires regional averaging.**

4 Evaluation of concentration data

I feel this chapter is not well integrated in the scope of the otherwise well-structured manuscript. Similar to Sec. 3.2.1 it also provides excessive (and seemingly unnecessary) mathematical derivations that seem unnecessary for the purpose of this manuscript. Also, while I am not an expert on $N_2$ geochemistry in surface waters, I would be surprised if these concepts and equations have not been presented and discussed in the existing literature.

**Reply: We agree that the equations in section 4.1 have already been discussed and presented in other works, which we have referenced. We will remove this part and focus solely on the new approach we developed for determining $N_2$ fixation rates in section 4.2. The main goal of this manuscript is to introduce a new method for determining $N_2$ fixation rates using GE-MIMS, based on concentration series obtained through long-term observations, preferably on a VOS as a measurement platform. This naturally involves not only the presentation of the measurement system but also how to process the acquired data to ultimately obtain the $N_2$ fixation rates. Therefore, we believe that this topic must be included into our manuscript.**

We have deleted section 4.1.

Lines 366-369

I don't agree. The physico-chemical properties of N2 are different to those of O2 and Ar, as demonstrated, for example, by the measurements in Sec. 3.2.2. These differences do result in fractionation of N2/Ar relative to O2/Ar.

**Reply: As we have mentioned above section 4.1 will be deleted in the revised manuscript.**

We have deleted section 4.1.

Section 4.2

The Schmidt-Number method provides a rough estimate of the gas exchange of the mixed layer with the atmosphere. However, if the Schmidt-Number model is really necessary here, I feel this discussion needs to be expanded with a quantitative assessment of the inherent uncertainties.

**Reply: We do not understand, what the reviewer means by Schmidt Number method or model. The Schmidt number, $Sc$, does not refer to any "estimate of the gas exchange". It is simply the dimensionless ratio between the kinematic viscosity and the diffusivity of the considered gas. It is related to the gas exchange transfer velocity $k$ by: $k \sim Sc^{-x}$ and used to convert $k$ which in chemical oceanography is usually referring to a standard Schmidt number of $Sc = 660$ ($CO_2$ at 25 °C in seawater at salinity of 35), $k_{660}$, to any other gas and temperature (e.g., Weber et al. (2018) and many others). For the exponent**

**$x$, a value of 1/2 is generally used at wind speeds above 3 m/s, but may increase at lower wind speeds (2/3).**
**In our case we use the change in the Ar concentration that must be driven by gas exchange to derive the $N_2$ gas exchange (implicitly we are determining $k$). This implies the use of the ratio $Sc(N_2)/Sc(Ar)$ which at a given temperature and salinity (viscosity) is given by the ratio of the corresponding diffusivities.**

Lines 418-420

As I understand it, the Schmidt-Number model breaks down at low wind speed, as gas exchange rates do not tend to zero without wind. This statement therefore warrants a more quantitative argument based on observed data.

**Reply: We refer to the reply above.**

Lines 424-425

This has been demonstrated with a GE-MIMS instrument in previous work [9].

**Reply: But Weber et al. used an entirely different approach to quantify the gas exchange.**

5. Conclusion

This chapter will need to be reworked to reflect to focus and new findings of a reworked manuscript.]

**Reply: We will rework the conclusions accordingly, including suggestions from the other reviewers.**

The revised conclusion now reads: "The results from our laboratory tests demonstrated that the GE-MIMS system is capable of directly determining cyanobacterial $N_2$ consumption and potentially also the associated $O_2$ production resulting from photosynthesis triggered by $N_2$ fixation. Ar concentrations for both the parametrization of air-sea gas exchange and for the reconstruction of abiotic background concentrations of biogeochemically active gases such as $N_2$ and $O_2$ could be measured with high precision and accuracy. Our measurement system is based on the same principle as that of Schmale et al. (2019), but it uses a membrane equilibrator, which in contrast to bubble/shower-type equilibrators, do not require ventilation and therefore constitutes a closed system. This ensures that the partial pressures in the gas phase are truly at equilibrium with the dissolved gases rather than merely at steady state (Schneider et al., 2007),
The individual components are designed to allow autonomous long term operation of the measurement system, particularly when installed on a VOS, such as that currently used for continuous $pCO_2$ measurements in the Baltic Sea (Gülzow et al., 2011; Schneider et al., 2014b; Jacobs et al. 2021). The resulting $N_2$, $O_2$ and Ar concentration time series will facilitate determinations of $N_2$ fixation rates and potentially NCP in selected regions of the Baltic Sea. The temporal dynamics of the above-mentioned biogeochemical processes can also be investigated. Furthermore, synchronous measurements of surface $N_2(Ar)$ and $pCO_2$, take advantage of both the direct determination of $N_2$ consumption by fixation and the high sensitivity of the $CO_2$ approach to production events (Schneider and Müller, 2018). The main limitations of existing approaches to quantifying $N_2$ fixation, which result from the analysis of

discrete samples and the use of the elemental composition of POM, are thus circumvented. Furthermore, the possibility of averaging over larger spatial scales due to the operation of the GE-MIMS on a VOS enhances its compatibility with process-based model results, which typically have a spatial resolution of several kilometers.".

References (Reviewer)

[1] Gas-equilibrium membrane inlet mass spectrometry with accurate quantification of dissolved-gas partial pressures (GE-MIMS-APP). Patent EP 4 109 092 A1, 2022.

[2] S A Ansari, S Chaudhury, P K Mohapatra, S K Aggarwal, and V K Manchanda. Recovery of plutonium from analytical laboratory waste using hollow fiber supported liquid membrane technique. Separation Science and Technology, 48(2):208–214, 2012.

[3] Matthias S Brennwald, A P Rinaldi, J Gisiger, A Zappone, and R Kipfer. Gas equilibrium membrane inlet mass spectrometry (GE-MIMS) for water at high pressure. Geoscientific Instrumentation, Methods and Data Systems, 13(1), 2024.

[4] Matthias S Brennwald, M Schmidt, J Oser, and R Kipfer. A portable and autonomous mass spectrometric system for on-site environmental gas analysis. Environmental Science and Technology, 50(24):13455–13463, 2016.

[5] Eliot Chatton, Thierry Labasque, Jérôme de La Bernardie, Nicolas Guihéneuf, Olivier Bour, and Luc Aquilina. Field continuous measurement of dissolved gases with a CF-MIMS: Applications to the physics and biogeochemistry of groundwater flow. Environmental Science and Technology, 51(2):846–854, 2017.

[6] G M Fryer. A theory of gas flow through capillary tubes. Proc. Royal Soc. London, Ser. A Math. Phys. Sci., 293(1434):329–341, 1966.

[7] S Giroud, Y Tomonaga, M S Brennwald, N Takahata, T Shibata, Y Sano, and R Kipfer. New experimental approaches enabling the continuous monitoring of gas species in hydrothermal fluids. Front. Water, 4, 2023.

[8] R P Schwarzenbach, P M Gschwend, and D M Imboden. Air-Water Exchange. Wiley, 2002.

[9] UW Weber, PG Cook, Matthias S Brennwald, R Kipfer, and TC Stieglitz. A novel approach to quantify air–water gas exchange in shallow surface waters using highresolution time series of dissolved atmospheric gases. Environmental Science and Technology, 53(3):1463–1470, 2018.

References (Authors)

Brennwald, M. S., Schmidt, M., Oser, J., and Kipfer, R.: A Portable and Autonomous Mass Spectrometric System for On-Site Environmental Gas Analysis, Environ. Sci. Technol., 50, 13455–13463, https://doi.org/10.1021/acs.est.6b03669, 2016.

Burlacot, A., Burlacot, F., Li-Beisson, Y., and Peltier, G.: Membrane Inlet Mass Spectrometry: A Powerful Tool for Algal Research, Front. Plant Sci., 11, https://doi.org/10.3389/fpls.2020.01302, 2020.

Cassar, N., Barnett, B. A., Bender, M. L., Kaiser, J., Hamme, R. C., and Tilbrook, B.: Continuous High-Frequency Dissolved O2/Ar Measurements by Equilibrator Inlet Mass Spectrometry, Anal. Chem., 81, 1855–1864, https://doi.org/10.1021/ac802300u, 2009.

data sheet: 3M$^{TM}$ Liqui-Cel$^{TM}$ MM-1.7x8.75 Series Membrane Contactor, Rev. 02, 2021, https://multimedia.3m.com/mws/media/1412495O/3m-liqui-cel-mm-1-7x8-75-series-membrane-contactor.pdf [20.09.2024]

Gülzow, W., Rehder, G., Schneider, B., Deimling, J. S. v., and Sadkowiak, B.: A new method for continuous measurement of methane and carbon dioxide in surface waters using off-axis integrated cavity output spectroscopy (ICOS): An example from the Baltic Sea, Limnol. Oceanogr. Methods, 9, 176–184, https://doi.org/10.4319/lom.2011.9.176, 2011.

Hoffmann, D. L., Richards, D. A., Elliott, T. R., Smart, P. L., Coath, C. D., and Hawkesworth, C. J.: Characterisation of secondary electron multiplier nonlinearity using MC-ICPMS, Int. J. Mass Spectrom., 244, 97–108, https://doi.org/10.1016/j.ijms.2005.05.003, 2005.

Khan, M. I., Lubner, S. D., Ogletree, D. F., and Dames, C.: Temperature dependence of secondary electron emission: A new route to nanoscale temperature measurement using scanning electron microscopy, J. Appl. Phys., 124, 195104, https://doi.org/10.1063/1.5050250, 2018.

Mächler, L., Brennwald, M. S., and Kipfer, R.: Membrane Inlet Mass Spectrometer for the Quasi-Continuous On-Site Analysis of Dissolved Gases in Groundwater, Environ. Sci. Technol., 46, 8288–8296, https://doi.org/10.1021/es3004409, 2012.

Manning, C. C., Stanley, R. H. R., and Lott, D. E. I.: Continuous Measurements of Dissolved Ne, Ar, Kr, and Xe Ratios with a Field-Deployable Gas Equilibration Mass Spectrometer, Anal. Chem., 88, 3040–3048, https://doi.org/10.1021/acs.analchem.5b03102, 2016.

Schmale, O., Karle, M., Glockzin, M., and Schneider, B.: Potential of Nitrogen/Argon Analysis in Surface Waters in the Examination of Areal Nitrogen Deficits Caused by Nitrogen Fixation, Environ. Sci. Technol., 53, 6869–6876, https://doi.org/10.1021/acs.est.8b06665, 2019.

Schneider, B., Sadkowiak, B., and Wachholz, F.: A new method for continuous measurements of O2 in surface water in combination with pCO2 measurements: Implications for gas phase equilibration, Mar. Chem., 103, 163–171, https://doi.org/10.1016/j.marchem.2006.07.002, 2007.

Schneider, B., Gülzow, W., Sadkowiak, B., and Rehder, G.: Detecting sinks and sources of CO2 and CH4 by ferrybox-based measurements in the Baltic Sea: Three case studies, J. Mar. Syst., 140, 13–25, https://doi.org/10.1016/j.jmarsys.2014.03.014, 2014.

Weber, U. W., Cook, P. G., Brennwald, M. S., Kipfer, R., and Stieglitz, T. C.: A Novel Approach To Quantify Air–Water Gas Exchange in Shallow Surface Waters Using High-Resolution Time Series of Dissolved Atmospheric Gases, Environ. Sci. Technol., 53, 1463–1470, https://doi.org/10.1021/acs.est.8b05318, 2019.

**Response to Reviewer 3 Comments (RC3):**

Summary

The technical note by Iwe and colleagues presents an analytical approach for the determination of $N_2$-fixation rates, which the authors envision as a tool for obtaining continuous, high-resolution measurements of $N_2$, Ar, and $O_2$. The main advantages of the proposed approach are that it improves the spatial and temporal coverage with respect to sporadic surveys (some of which use discrete sampling methods), and that it would enable the users to conduct detailed assessments of net community production (NCP) in surface waters of different oceanic regions, while accounting for small-scale variability.

General assessment

Strengths: Given the pivotal role of $N_2$-fixation, the topic of this technical note is certainly relevant. Although the principles of the individual methods (gas equilibration and MIMS) have been used in other studies for similar applications, their combination and optimization for underway measurements is novel. Besides the obvious advantages of being able to derive N2-fixation rates and NCP over large areas and with potentially unprecedented temporal coverage, this approach might enable a better understanding of carbon and nitrogen dynamics in surface waters. Overall, the manuscript is well written, the approach followed is clear and the specific aims (1. Assessing equilibration times and full equilibrium; and 2. Assessing the system's performance in terms of precision, accuracy, limits of detection) are adequately addressed and substantiated with laboratory-based experiments.

Weaknesses: The major drawback of this contribution is that the authors present it in a way that it has been optimized for surveys on board voluntary observing ships (VOS), without providing data/experiments derived from an at-sea deployment. As it stands, the manuscript shows an assessment that the system is, in principle, capable of conducting measurements on such a vessel just as much as it could do in any other type of application. Beyond this, perhaps semantic issue, there are practical considerations that need to be accounted for when systems are installed in an unattended manner (as I am sure it is known to some of the coauthors). These include strong temperature variability (potentially affecting both hardware and software), potential contamination, vibration, biofouling, etc. Because of this, several parts of the text (starting with the title) can be considered misleading in the absence of direct evidence.

Overall, it is my opinion that this is a contribution worthy of being published after some issues are addressed. I would be reluctant to ask the authors for data from an at-sea deployment at this stage, but my recommendation would be to reformulate so that it is clear that their approach paves the way for further studies that do carry out the deployments on VOS.

**Reply: We would like to thank the reviewer for acknowledge the value of our work and hope that the changes we applied will help to clarify the concerns. Among other things, we will present a more detailed description of the practical implementation of the measurement system on a voluntary observing ship (VOS).**

Specific comments:

Throughout the text: I spotted a few format inconsistencies with the usage of chemical names (e.g. sometimes "$O_2$", sometimes "oxygen", and also not all subscripts are correct).

**Reply: These will be corrected in the revised manuscript.**

We have resolved the inconsistencies in our revised manuscript.

l. 1 – 3 (Title):

This approach can, in principle be applied to any survey type and in this manuscript no data from VOS is shown. I would therefore include this as a potentially useful application in the context of long-term observatories.

**Reply: We will clarify in the revised manuscript that our approach is primarily intended for use on voluntary observing ships (VOS), while also acknowledging its applicability to other survey types, particularly in the context of long-term observations. We have also revised the title to reflect this broader applicability.**

The new title now reads: "Technical Note: Testing a new approach for the determination of $N_2$ fixation rates by coupling a membrane equilibrator to a mass spectrometer for long term observations".

l. 18 – 19:

("The GE-MIMS is designed for…"): Perhaps the authors could describe this as a "proof-of-concept" in view of its future application to conduct observations in VOS.

**Reply: We will emphasize that our measurement system is suitable for long-term observations in general using various platforms. However, it will be made more clear that VOS or similar platforms are essential for achieving the temporal and spatial resolution required for our approach, when investigating areas such as the Baltic Sea.**

We changed the sentence accordingly: "Our GE-MIMS approach is designed for long-term observations on various platforms such as voluntary observing ships (VOS). The latter are particularly suited to achieve the temporal and spatial resolution necessary for studying large scale $N_2$ fixation in regions such as the Baltic Sea.".

l. 84:

"provide" instead of "provides"

**Reply: This will be changed.**

The "s" was removed accordingly.

l. 99 (Figure 1 caption):

To me most abbreviations were clear, but there might be readers not yet familiar with this kind of analytical setup. Therefore I would recommend the authors to include abbreviations also here (I noticed that they are used in the text, which is good, but some are far from the actual figure).

**Reply: We appreciate the feedback and will ensure that all abbreviations are clearly defined in the caption.**

We have revised the Figure 1 caption accordingly: **"**Figure 1: Schematic diagram of the gas equilibrium – membrane-inlet mass spectrometry (GE-MIMS) system. V1: 2-position valve (computer controlled), V2,3,4: solenoid valves (computer controlled), V5,6: valves, T1: temperature probe, P1,2: pressure sensors, PP: perestaltic pump, FM: flow meter, DG: degassing cylinder.".

l. 103 - 104 ("A pressure gauge (P2) was installed"):

I was wondering whether the authors could add some values (or an empirical threshold) here. This would be good both to ensure repeatability and also guide potential new users of this approach.

**Reply: We will include a recommendation in the text, saying that for our setup the filter has to be replaced at 1 bar overpressure to ensure the safety and reliability of the system. This is a value we determined in laboratory tests using our method.**

We have inserted the following recommendation in line 104: "For our setup, the filter cartridge had to be replaced at 1 bar overpressure to ensure system safety and reliability.".

l. 113 – 115:

Virtually unattended deployment in a VOS will require a suitable alternative. I am guessing the authors might be able to provide useful suggestions on this.

**Reply: We agree with the reviewer and will clarify in the manuscript that other alternatives to the Liquicel membrane other than Permselect were not investigated.**

We have changed the sentence accordingly: "In addition to the Liqui-Cel membrane, we tested a membrane equilibrator produced by PermSelect (PDMSXA-1.0), in which the gas exchange between the water and gas phase is mediated by dense hollow fibers consisting of polydimethylsiloxane (PDMS).".

l. 150 – 154:

This information could be conveyed more clearly with a graph (e.g. an Allan plot).

**Reply: We will include an Allan plot to discuss the results more clearly in our revised manuscript.**

The Allan plot in Figure 1R shows the Allan deviation ($\sigma_A$) across 60 pre-averaged measurements (refer to Sec. 2.2) using two detectors (Faraday Cup and SEM) for $N_2$. Each data point represents the variation as more averaged measurements are combined over increasing time intervals ($\tau$, $\tau = 360$ s between two measurements). Although periodic calibration was performed approximately every 30 minutes, the observed minima in the Faraday Cup's curve likely stem from other factors, such as transient stability improvements. Overall, the plot indicates that deviation increases as more measurements are averaged together, with the Faraday Cup showing lower deviations than the SEM due to its reduced baseline noise.

Since the original purpose of creating an Allan plot (to determine an appropriate integration time $\tau$, where noise and drift behavior produce the smallest standard deviations) is not applicable here - given that we are not considering a long-term measurement but rather 60 measurements averaged from 100 data points each, with calibration performed in between - we have decided to not include the plot in the manuscript. All the information that could be derived from the Allan plot are already presented in Table 1, such as the smaller standard deviation when using the Faraday Cup.

[Figure]

**Figure 1R: Allan deviation plot of $N_2$ measurements using a Faraday Cup (blue) and a Secondary Electron Multiplier (SEM, black) detectors. Each point represents the Allan deviation as a function of averaging time $\tau$, calculated over 60 pre-averaged measurement cycles, each separated by 360 seconds.**

l. 215 – 249:

The full mathematical derivation is not novel and it seems unnecessary in this part of the manuscript. I would suggest the authors to shift this to an appendix.

**Reply: We will move this part to the Appendix.**

This is now part of Appendix C: "To calculate the transfer coefficient, $k_n$, we first derive an equation for the equilibration time, $\tau_{nf}$, for the hypothetical case in which there is no water flow (see Sec. 3.2.1). The flux across the membrane is driven by the partial pressure difference according to the general flux equation [Eq. (35)]:

$$\frac{\partial n_g}{\partial t \cdot A} = -k_n \cdot \Delta p \,, \tag{35}$$

with:
$\frac{\partial n}{\partial t}$ – change with time of the moles of a gas in the gas side of the equilibrator [mole $\cdot$ s$^{-1}$]
$A$ – membrane area [m$^2$]
$k_n$ – mass (mole) transfer constant [mol $\cdot$ s$^{-1}$ $\cdot$ m$^{-2}$ $\cdot$ atm$^{-1}$]
$\Delta p$ – partial pressure difference: $p_g$ - $p_w$ [atm]
subscript g refers to the gas side of the membrane equilibrator and w to the water side

Using the ideal gas law, $\partial n_g$ is replaced by $\partial p_g$ according to Eq. (36):

$$\frac{\partial p_g}{\partial t} = \frac{-k_n \cdot A \cdot R \cdot T}{V_g} \cdot \Delta p \,, \tag{36}$$

with:

$\partial p$ - change in the partial pressure of a gas [atm]
$R$ – universal gas constant [$m^3 \cdot atm \cdot mol^{-1} \cdot K^{-1}$]
$T$ – absolute Temperature [K]
$V$ – volume [$m^3$]

To describe $\Delta p$ only as a function of $p_g$, the total moles (gas side + water side) of the considered gas, $n_t$, which is constant at zero flow, is introduced, as shown in Eq. (37):

$$n_t = V_g \cdot \frac{p_g}{R \cdot T} + V_w \cdot p_w \cdot s \,, \tag{37}$$

with:

$s$ – solubility constant [$mol \cdot m^{-3} \cdot atm^{-1}$]

$p_w$ is thus given as shown in Eq. (38):

$$p_w = \frac{n_t - \left(\frac{p_g}{R \cdot T}\right) \cdot V_g}{V_w \cdot s} \,, \tag{38}$$

and $\Delta p$ is expressed using Eq. (39):

$$\Delta p = p_g - \frac{n_t - \left(\frac{p_g}{R \cdot T}\right) \cdot V_g}{V_w \cdot s} \,, \tag{39}$$

The differentiation of Eq. (39) yields Eq. (40):

$$d(\Delta p) = \left(1 + \frac{V_g}{R \cdot T \cdot V_w \cdot s}\right) dp_g \,, \tag{40}$$

Replacing $\partial p_g$ in Eq. (36) then yields Eq. (41):

$$\frac{d(\Delta p)}{dt} = -k_n \cdot A \cdot \left(\frac{R \cdot T}{V_g} + \frac{1}{s \cdot V_w}\right) \cdot \Delta p \,, \tag{41}$$

The integration of which provides an exponential equation [Eq. (42)]:

$$\Delta p = \Delta p_0 \cdot \exp\left[-k_n \cdot A \cdot \left(\frac{R \cdot T}{V_g} + \frac{1}{s \cdot V_w}\right) \cdot t\right] \,, \tag{42}$$

with a time constant [$s^{-1}$] that equals the reciprocal equilibration time $\frac{1}{\tau_{nf}}$ (no water flow), resulting in Eq. (43) and Eq. (44):

$$\Delta p = \Delta p_0 \cdot \exp\left(\frac{-t}{\tau_{nf}}\right) \,, \tag{43}$$

$$\tau_{nf} = \frac{1}{k_n \cdot A \cdot \left(\frac{R \cdot T}{V_g} + \frac{1}{V_w \cdot s}\right)} \,, \tag{44}$$

In addition to the geometric dimensions ($V_g = 1.40 \cdot 10^{-6}$ $m^3$, $A = 0.92$ $m^2$) of the membrane equilibrator and the thermodynamic properties, the gas exchange and thus the equilibration time is controlled by the transfer coefficient $k_n$. The latter can be calculated, using the experimentally determined equilibration times ($\tau$ ($N_2$) = 288 s, Sec. 3.2.2). Since these were determined with a water flow, we assume that $V_w$ is infinitely large, thereby modifying Eq. (44) to yield Eq. (45) and thus $k_n$ for $N_2$:

$$k_n = \frac{V_g}{\tau \cdot A \cdot R \cdot T} \,, \tag{45}$$

$k_n = 2.21 \cdot 10^{-5}$ $mol \cdot m^{-2} \cdot s^{-1} \cdot atm^{-1}$.".

l. 258 – 260:

It is hard to grasp how the underlying assumption of no water flow could be directly applied to operation conditions in which indeed there will be seawater flowing through the system. In my opinion this needs further explanation.

**Reply: The purpose of using a model that examines gas exchange with stagnant water was to provide a simplified representation for understanding the fundamental principles of gas dynamics. This approach was intended to help readers, especially those less familiar with the topic, grasp the essential mechanisms at play. However, in response to the reviewer comments, we have moved this model to the appendix to clarify its role and significance in our overall analysis.**

See Appendix C.

l. 382 ("(…) denitrification in deep waters."):

A citation seems to be missing here.

**Reply: We will remove the section 4.1 since the focus of our manuscript is on the determination of the $N_2$ fixation and its importance for the surface water N budget.**

We have deleted section 4.1.

l. 394 ("Ignoring vertical mixing (…)"):

This choice should be substantiated.

**Reply: We will provide a more detailed explanation to substantiate this choice in the revised manuscript, by describing the surface stratification of the Baltic Sea during mid-summer and especially during periods that favor the development of cyanobacteria blooms, which indicate that vertical mixing might be ignored.**

We have revised the sentence and added the following information: "Any change in the $N_2$ concentration can be described as the effect of $N_2$ fixation and $N_2$ gas exchange with the atmosphere (Eq. 20) if vertical mixing across the thermocline is ignored. The latter is justified since $N_2$ fixation typically takes place during low wind speeds (< 5 m/s) which lead to a rising thermocline and warming of the surface layer (up to 22 °C) (Müller et al., 2021).".

l. 434 ("(…) such that also currently used for continuous pCO2 measurements (…)"):

A citation seems to be missing here.

**Reply: We will add the missing reference (e.g., Schneider et al., 2014 and many others).**

We have added the missing references and the sentence now reads: "The individual components are designed to allow autonomous long term operation of the measurement system, particularly when installed on a VOS, such as that currently used for continuous $pCO_2$ measurements in the Baltic Sea (Gülzow et al., 2011; Schneider et al., 2014b; Jacobs et al. 2021).".

l. 435 ("(…) will facilitate determinations of NCP"):

A further potential application of the approach presented by the authors would be to combine it with underway measurements of $N_2O$, since this might help further constraining uncertainties in $O_2/Ar$ based NCP estimates (see Cassar et al., GRL, 8961–8970, 2014).

**Reply: We thank the reviewer for the suggestion, which will be taken into account for future applications.**

References (Authors)

Schneider, B., Gülzow, W., Sadkowiak, B., and Rehder, G.: Detecting sinks and sources of CO2 and CH4 by ferrybox-based measurements in the Baltic Sea: Three case studies, J. Mar. Syst., 140, 13–25, https://doi.org/10.1016/j.jmarsys.2014.03.014, 2014.

---

## Referee Report (RR1)

**Review of EGUSPHERE-2024-2049-revised Manuscript**

**Overview and general comments**

I have reviewed this manuscript before, and I will focus on my previous review comments in this re-review as discussed below. The authors adressed some of the comments in my initial review, but I still have reservations about the overall quality of the work as described in my updated list of my initial review comments below.

Overall, my main concerns are:

- The description of the gas-equilibration model was moved from the main text to the appendix, but the model itself was not modified substantially and still suffers from the same physical misconceptions.

- The description and discussion of the data processing and uncertainty assessment are not fully adequate for a methods paper.

- The work uses design concepts that are covered / protected by a patent. I believe it would be wise to clarify the rights to use these concepts before moving forward.

I recommend to resolve these points before moving forward.

**Details and specific comments**

This is a list of my initial review comments with follow-up comments related to the revised manuscript (*in italics*). I removed the comments that were sorted out by the revision from this list.

**1. Introduction**

Line 67 The dynamic steady state in a conventional MIMS is controlled by many more factors than just the dissolved gas concentrations and the MS pumping rate. The water flow rate, the geometry of the membrane system, water salinity, temperature, aging of the membrane material and its gas permeation properties, etc. play a crucial role.
*The authors' revision fails the point. My comment was aimed at discussing the numerous parameters that control the operation and measurement results (and their uncertainties!) of the MIMS technique.*

**2.1 Membrane equilibrator**

Appendix A, line 117/118  Using a pressure sensor to determine the total gas pressure and to quantify the partial pressures of the different gas species in the membrane equilibrator has been previously described in patent EP 4 109 092 A1, which should be referenced here. *The authors' system clearly **does** use a pressure sensor to determine the total gas pressure, which in turn is needed to quantify the partial pressures of the various gas species. I'd advise to consult an expert to avoid any potential for patent conflicts.*

Line 112  How "negligible" is the gas removal? This is a crucial control for the accuracy of the analytical results and calls for a quantitative argument. *The authors argue that this is "...a minor effect and applies to the calibration and measurements as well." However, the effect does **not** apply to the calibration measurements because the calibration gas it not taken from the membrane module.*

Line 114 and 115  Why would a clogged capillary pose a risk for the MS? I'd rather argue that the clogging protects the MS from accidents with too much water. *The authors now argue that the clogging of the pores of the membrane material by particles could be an issue. This has nothing to do with water in the capillary. I don't understand and am unsure about this revision.*

Line 116 and 117  Is this a confusion between accuracy and precision? *The authors revised this to indicate the precision of the temperature sensor, but I think the information on accuracy would be much more relevant, because it is needed to determine the dry-gas pressure.*

Line 121 and 122  Why would the depressurization in the outflow tubing have an effect on the gas/water equilibrium in the membrane module? Please explain. *The authors' response makes sense, but they did not add their explanation to the revised manuscript.*

**2.2 Mass spectrometry**

Line 139/140  The Faraday cup and SEM are likely used not only for detection, but rather for quantification. *It seems the authors did not mention the SEM in their revision.*

Line 140–142  One might expect a better signal/noise ratio from the SEM, in contrast to the observation reported here. Why is this? Please elaborate. *I agree with the authors' response, but it would still be useful to add a comment to the manuscript to avoid confusion with readers.*

Line 143/144 Quantification of the partial pressures must be based on the peak heights in the mass spectrum. To determine the peak heights, the baseline values therefore need to be subtracted from the peak-top values measured at the indicated m/z positions. Were the baseline values measured? At which m/z values?

*At $m/z = 3$ the signal is governed by the peak of H-3 molecules in the mass spectrometer, so this does not seem like a suitable position in the spectrum to determine the base line.*

Lines 150–154 Why 60 repetitions for averaging? Why a 6 h long test period? The usual approach is to optimize the signal/noise ratio while minimizing the effect of drift. This is commonly done using an Allan plot. Is this what the authors did? Please explain.

*The Allan plot seems incomplete, as it only shows the increasing trend of the Allan deviation corresponding to drift. The plot is missing the short-term range where the Allan deviation shows a decreasing trend corresponding to random noise. It is therefore impossible to determine the optimal integration time corresponding to the minimum Allan deviation at the transition between these two regimes (random noise vs. drift). I believe the the averaging/integration chosen by the authors is not optimal.*

**3.1 Accuracy and Precision**

Line 168/169 Estimating the water vapor pressure by assuming saturation in the GE-MIMS equilibrator has been described in patent EP 4 109 092 A1, which should be referenced here.

*I'd recommend to consult an expert to avoid any potential for patent conflicts.*

Lines 173–180 Using Henry's Law to convert the partial pressures to dissolved gas concentrations has been described in previous GE-MIMS work, which should be referenced here (see previous comments).

*My comment is aimed at referencing previous GE-MIMS work, not Henrys Law.*

Lines 182–188 Air-equilibrated water (AEW) is a good reference to assess the analytical performance, but fabrication of AEW is notoriously difficult. I would recommend comparison and validation of their GE-MIMS system with other (validated and established) methods for dissolved-gas quantification.

*The authors claim the "stability of the measurement values shown in Figure 2". This indicates the precision of their measurements, not the **accuracy** of their AEW reference used to assess the analytical performance.*

192/193 The RSD is normalized relative to the concentration value. A lower concentration value should therefore not result in a lower RSD.

*The authors seem to agree, but did not revise the manuscript accordingly.*

**3.2.1 Theory of equilibration kinetics**

I am not convinced that this section adds much value to the manuscript. On the one hand, it assumes that the water is stagnant inside the membrane module (it is not), and it assumes that the membrane provides the bottleneck for the gas transfer between the water the gas phase. However, the resistance of the membrane material to the gas transfer is marginal (the authors can convince themselves about this by blowing into the water inlet of a dry module while blocking the water outlet, and observe how the air easily escapes through the membrane material into the gas headspace). In contrast, the main bottleneck for the transfer of gas species between the water and the gas headspace is expected to result from the gas exchange mechanisms at the gas/water interface (see for example [**?**]). The main outcome of section 3.2.1 is that the partial-pressure equilibration follows an exponential function, which comes to no surprise given the assumption of a first-order exchange kinetic, and which does not warrant any mathematical derivation. A second result is equation (27), which provides a formula to calculate the equilibration time. However, this equation relies on incorrect model assumptions (stagnant water, membrane as bottleneck for gas/water transfer) and therefore does not provide much insight.

*The authors (i) moved the model to the appendix and they (ii) argued that gas exchange was modeled assuming stagnant water. I do not understand how these points would be related to my comment, or how the revision would improve the manuscript.*

Lines 285–286 This seems like a trivial finding since the removal of gas from a finite, stagnant volume of water will result in a lower dissolved-gas concentration, and hence in a lower partial pressure at equilibrium. In reality, there's a continuous flow of water through the membrane module, which means there's a (virtually) infinite amount of water available for equilibration with the gas headspace. Again, this shows that the model concept and equations are based on inappropriate assumptions.

*The authors argue that "the purpose of using a model that examines gas exchange with stagnant water was to provide a simplified representation for understanding the fundamental principles of gas dynamics. This approach was intended to help readers, especially those less familiar with the topic, grasp the essential mechanisms at play." However, the stagnant water assumption does **not** represent the mechanisms at play and also makes the mathematical treatment overly complicated. Assuming a high flow of (non-stagnant) water with defined (constant) dissolved-gas concentrations would reflect the true GE-MIMS mechanisms much better and would make for much simpler maths.*

**3.2.2 Measurement of $\tau$**

This section provides robust information on the time needed to attain gas/water equilibrium in the membrane module, and provides a useful basis to estimate the spatial resolution of the dissolved-gas data recorded on a moving ship. The measured equilibration times $\tau$ are approximately 50 % higher than those calculated from the model in Sec. 3.2.1, which supports my impression that the model is inaccurate and seems inappropriate to optimize the operation of the GE-MIMS method for the dissolved gas monitoring described in the manuscript. To this end, the experimentally determined $\tau$ values are more suitable, and the model could be removed from the manuscript entirely. *The authors added a note to the manuscript acknowledging the discrepancy of the observed $\tau$ relative to those expected from their model, but they did not attempt to revise (or remove) their inadequate model.*

Lines 340  I don't see the need for 29 equations simply to state that the partial pressures evolve exponentially towards their equilibrium value. This seems like a trivial result of the assumed first-order gas-exchange kinetic. *The authors argue that the extensive set of equations is necessary to provide the theoretical background of their model. However, the theoretical background is based on a numer of physical misconceptions as described previously.*

Fig. 4  The right panel seems unnecessary, as it shows the same data as the one on the left. I'd suggest to show only the left panel and add the fitted exponential curve. *The authors argue that the two panels do not show the same data, but at the same time they also indicate that the right panel presents part of the data in the left panel – so it* **is** *the same data. I still think this figure could be simplified a lot.*

Lines 349–351  The ratios of the measured and modeled $\tau$ values are 4.8/4.3 = 1.1 ($N_2$), 3.2/2.2 = 1.5 ($O_2$), and 3.0/2.0 = 1.5 (Ar). In other words, the true (measured) values are up to 50 % higher than those estimated from the model. I don't see how this large discrepancy can be explained by non-ideality of the gas or "impurities" of the membrane. As mentioned before, there are more fundamental flaws in model assumptions. *The observed data simply show that the model is not adequate (as pointed out before).*

Lines 357–360  The membrane module used in this work is rather large and therefore exhibits long equilibration times of 12–20 min. Why did the authors not use much smaller membrane modules that would allow equilibration within about 3 min [**?**], which would in turn also provide approximately 5× better spatial resolution in their dissolved-gas monitoring? *The authors argue that the large size of the volume helps to prevent disturbing the solubility equilibrium. However, the equilibrium disturbance of the headspace*

*gases is mainly controlled by the dynamic balance of (i) the gas consumption to the mass spectrometer and (ii) the re-supply of gas from the water. Again, the authors' conceptual model of how the GE-MIMS technique works seems incomplete.*

Lines 424–425 This has been demonstrated with a GE-MIMS instrument in previous work (Weber et al.).

*The Weber et al. work **did** use the GE-MIMS technique to determine the gas exchange, and I'd suggest to acknowledge that.*